# African-specific molecular taxonomy of prostate cancer

Weerachai Jaratlerdsiri[1,2], Jue Jiang[1,2], Tingting Gong[1,2,15], Sean M. Patrick[3], Cali Willet[4], Tracy Chew[4], Ruth J. Lyons[2], Anne-Maree Haynes[2], Gabriela Pasqualim[5,6], Melanie Louw[7], James G. Kench[8], Raymond Campbell[9], Lisa G. Horvath[2,10], Eva K. F. Chan[2,16], David C. Wedge[11], Rosemarie Sadsad[4], Ilma Simoni Brum[5], Shingai B. A. Mutambirwa[12], Phillip D. Stricker[2,13], M. S. Riana Bornman[3] & Vanessa M. Hayes[1,2,3,14] ✉

Prostate cancer is characterized by considerable geo-ethnic disparity. African ancestry is a significant risk factor, with mortality rates across sub-Saharan Africa of 2.7-fold higher than global averages[1]. The contributing genetic and non-genetic factors, and associated mutational processes, are unknown[2,3]. Here, through whole-genome sequencing of treatment-naive prostate cancer samples from 183 ancestrally (African versus European) and globally distinct patients, we generate a large cancer genomics resource for sub-Saharan Africa, identifying around 2 million somatic variants. Significant African-ancestry-specific findings include an elevated tumour mutational burden, increased percentage of genome alteration, a greater number of predicted damaging mutations and a higher total of mutational signatures, and the driver genes *NCOA2*, *STK19*, *DDX11L1*, *PCAT1* and *SETBP1*. Examining all somatic mutational types, we describe a molecular taxonomy for prostate cancer differentiated by ancestry and defined as global mutational subtypes (GMS). By further including Chinese Asian data, we confirm that GMS-B (copy-number gain) and GMS-D (mutationally noisy) are specific to African populations, GMS-A (mutationally quiet) is universal (all ethnicities) and the African–European-restricted subtype GMS-C (copy-number losses) predicts poor clinical outcomes. In addition to the clinical benefit of including individuals of African ancestry, our GMS subtypes reveal different evolutionary trajectories and mutational processes suggesting that both common genetic and environmental factors contribute to the disparity between ethnicities. Analogous to gene–environment interaction—defined here as a different effect of an environmental surrounding in people with different ancestries or vice versa—we anticipate that GMS subtypes act as a proxy for intrinsic and extrinsic mutational processes in cancers, promoting global inclusion in landmark studies.

Prostate cancer is a common heterogeneous disease that is responsible annually for more than 1,400,000 new diagnoses and 375,000 male-associated deaths worldwide[1]. Characterized by a highly variable natural history and diverse clinical behaviours[4], it is not surprising that genome profiling has revealed extensive intra- and intertumour heterogeneity and complexity[5,6]. The identification of oncogenic subtypes[7] and actionable drug targets[8] are moving prostate cancer management a step closer to the promise of precision medicine[7,9–12]. Although high-income European ancestral countries are well along the road to incorporating cancer genomics in all aspects of cancer care[13], the rest of the world lags behind, with a notable absence in sub-Saharan Africa[14]. Prostate cancer is no different, with a single large-scale study out of China[11]; in 2018, we provided a snapshot for sub-Saharan Africa, reporting an elevated mutational density in a mere six cases[15]. With mortality rates of greater than double compared with high-income countries and quadrupled for greater Asia, in sub-Saharan Africa, prostate cancer is the top-ranked male-associated cancer both by diagnosis and deaths, including southern Africa with age-standardized rates of 65.9 and 22

[1]Ancestry and Health Genomics Laboratory, Charles Perkins Centre, School of Medical Sciences, Faculty of Medicine and Health, University of Sydney, Camperdown, New South Wales, Australia. [2]Genomics and Epigenetic Theme, Garvan Institute of Medical Research, Darlinghurst, New South Wales, Australia. [3]School of Health Systems & Public Health, University of Pretoria, Pretoria, South Africa. [4]Sydney Informatics Hub, University of Sydney, Darlington, New South Wales, Australia. [5]Endocrine and Tumor Molecular Biology Laboratory (LABIMET), Instituto de Ciências Básicas da Saúde, Universidade Federal do Rio Grande do Sul, Porto Alegre, Brazil. [6]Laboratory of Genetics, Instituto de Ciências Biológicas, Universidade Federal do Rio Grande, Rio Grande, Brazil. [7]National Health Laboratory Services, Johannesburg, South Africa. [8]Department of Tissue Pathology and Diagnostic Oncology, Royal Prince Alfred Hospital and Central Clinical School, University of Sydney, Sydney, New South Wales, Australia. [9]Kalafong Academic Hospital, Pretoria, South Africa. [10]Medical Oncology, Chris O'Brien Lifehouse, Royal Prince Alfred Hospital and Faculty of Medicine and Health, University of Sydney, Camperdown, New South Wales, Australia. [11]Division of Cancer Sciences, University of Manchester, Manchester, UK. [12]Department of Urology, Sefako Makgatho Health Science University, Dr George Mukhari Academic Hospital, Medunsa, South Africa. [13]Department of Urology, St Vincent's Hospital, Darlinghurst, New South Wales, Australia. [14]Faculty of Health Sciences, University of Limpopo, Mankweng, South Africa. [15]Present address: Human Phenome Institute, Fudan University, Shanghai, China. [16]Present address: NSW Health Pathology, Sydney, New South Wales, Australia. ✉e-mail: vanessa.hayes@sydney.edu.au

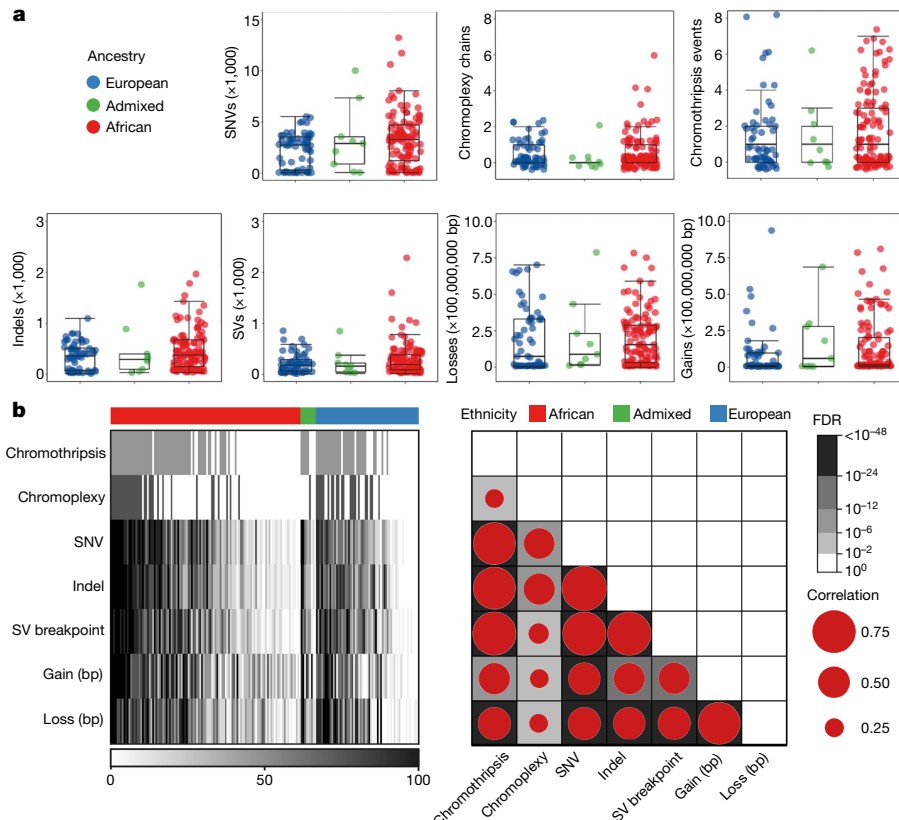

**Fig. 1 | Mutational density in prostate tumours of individuals with different ancestries. a**, The distribution of somatic aberrations (event number or number of base pairs) for 7 mutational types across 183 tumour–blood WGS pairs representing $n = 61$ European, $n = 113$ African and $n = 9$ admixed individuals. The box plots show the median (centre line), the 25th and 75th percentiles (box limits), and ±1.5× the interquartile range (whiskers).

**b**, The different types of mutational burden observed in this cohort. The samples were percentile-ranked and then ordered on the basis of the sum of percentiles across the mutational types observed in each ancestral group (left). Right, Spearman correlation is shown between mutational types, with the dot size representing the magnitude of correlation and the background colour giving the statistical significance of FDR values.

per 100,000, respectively[1]. Through the Southern African Prostate Cancer Study (SAPCS), we report a 2.1-fold increase in aggressive disease (grades 4–5) and 4.8-fold increase in prostate-specific antigen levels at diagnosis compared with African Americans[16].

Here we describe, to our knowledge, the largest cancer and prostate cancer genomics data for sub-Saharan Africa, including 123 South African men. Controlling for study artefacts, an additional 53 Australian and 7 Brazilian individuals were passed simultaneously through the same high-depth whole-genome sequencing (WGS), mutation-calling and analytical framework. Focusing on treatment-naive cases (100% South Africans, 98% Australians and two confirmed Brazilians) and aggressive tumours (grades 4–5 for 72.2% South Africans, 86.8% Australians and 85.7% Brazilians; Extended Data Fig. 1a) at biopsy (100% South Africans) or surgery (100% Australians, 62.5% Brazilians) and patient-matched blood achieving coverages of 88.69 ± 14.78 and 44.34 ± 8.11, respectively (median ± s.d.; Supplementary Table 1), we uniformly generated, called and assessed about 2 million somatic variants. Through ancestral classification (genetic ancestry over self-identified ethnicity), we show a greater number of acquired genetic alterations within African individuals while identifying both globally relevant and African-specific genomic subtypes. Combining our somatic variant dataset with that published for ethnically defined European[7,8,17,18] and Chinese[11] prostate cancer genomes, we reveal a prostate cancer taxonomy with different clinical outcomes. The inclusion of 2,658 cancer genomes from the ICGC/TCGA Pan-Cancer Analysis of Whole Genomes (PCAWG)[13] expanded our global mutational subtyping between cancer types. Using known clock-like mutational processes in each subtype, we inferred mutation timing of oncogenic drivers in broad periods of

tumour evolution and calculated the mutation rates for each subtype that had a distinctive tumour evolution pattern. Combined, these analyses enable us to demonstrate how global inclusion in cancer genomics can unravel unseen heterogeneity in prostate cancer in terms of its genomic and clinical behaviours.

## Genetic ancestry

Genetic ancestries were estimated for the 183 patient donors using a joint dataset in a unified analysis aggregated from a collection of geographically matched African ($n = 64$) and European ($n = 4$) deep-coverage published and unpublished reference genomes[19]. Ancestries were assigned using 7,472,833 markers as African ($n = 113$, all South Africans), with greater than 98% contribution; European ($n = 61$; 53 Australians, 5 South Africans and 3 Brazilians), allowing for up to 10% Asian contribution (with a single outlier of 26%); and African–European admixed ($n = 9$; 5 South Africans and 4 Brazilians), with as little as 4% African or European contribution (Extended Data Fig. 1b).

## Total somatic mutations

In 183 prostate tumours, we identified 1,067,885 single-nucleotide variants (SNVs), 11,259 dinucleotides, 307,263 small insertions and deletions (indels, <50 bp), 419,920 copy-number alterations (CNAs) and 22,919 structural variants (SVs), with each mutational type elevated in tumours from African individuals (Fig. 1a). A median of 37.54% ± 5.51 of SNVs were C-to-T mutations, and the transition and transversion ratio was 1.282 cohort-wise. Tumours from African individuals had

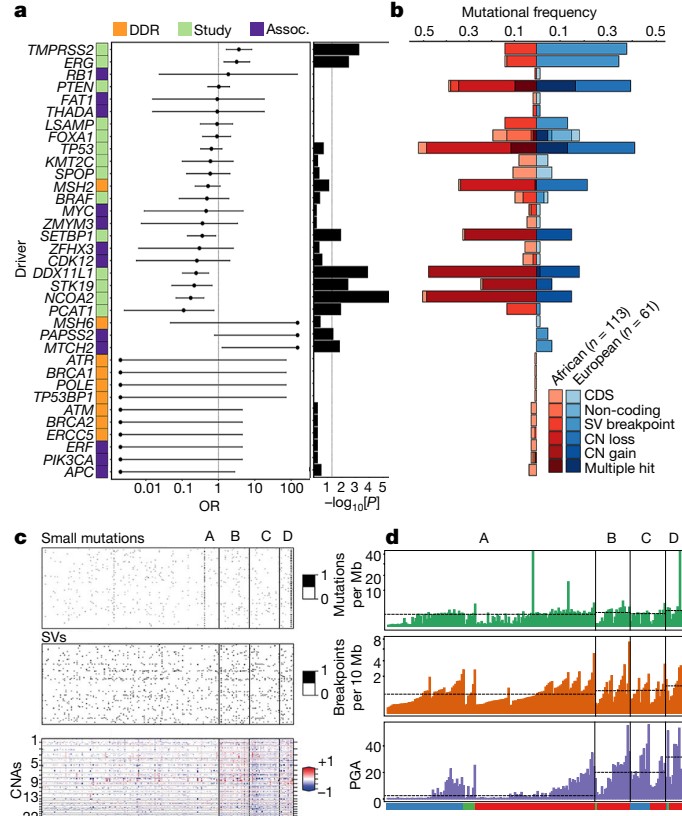

**Fig. 2 | Taxonomy and differences in driver mutations in prostate cancer by ancestry. a**, The selected 35 driver genes classified as (1) the most altered in this study (>10 patients), irrespective of ancestry (green); (2) DNA-damage repair (DDR) genes that are known to be associated with African ancestry (orange); (3) other ancestry-associated genes studied in prostate cancer (assoc., purple). The OR, 95% confidence interval and two-sided $P$ value (<0.05) were calculated using Fisher exact tests for count data and including 10 African-specific (OR = 0) and 3 European-specific (OR = infinity) genes. Significance was observed for *TMPRSS2* ($P = 0.0006$), *ERG* ($P = 0.003$), *SETBP1* ($P = 0.012$), *DDX11L1* ($P = 0.0001$), *STK19* ($P = 0.004$), *NCOA2* ($P = 3.14 \times 10^{-6}$), *PCAT1* ($P = 0.012$), *PAPSS2* ($P = 0.042$) and *MTCH2* ($P = 0.014$). **b**, The mutational frequency of the altered driver genes between Africans and Europeans by mutational type (CDS, non-coding, SV and CNA). **c**, An integrative clustering analysis reveals four distinct molecular subtypes of prostate cancer. The molecular subtypes are illustrated by small somatic mutations (coding regions and non-coding elements), somatic CNAs and somatic SVs. The proportion and association between the iCluster membership and patient ancestry are illustrated in **d**. Additional unsupervised consensus clustering on each data type was performed and mostly recapitulated the subtypes by integrative analysis. **d**, Total somatic mutations across four molecular subtypes in this study. The dashed lines indicate the median values of mutational densities across the four subtypes. For each subtype, patients are ordered on the basis of their ancestry.

a higher rate of small mutations (SNVs and indels), with a median of 1.197 mutations per Mb (range 0.031–170.445) compared with those of Europeans (1.061 mutations per Mb; $P = 0.013$, two-sample $t$-test; exclusion of hypermutated tumours at >30 mutations per Mb, $P = 0.028$). The percentage of genome alteration (PGA) was similarly greater in Africans (7.26% versus 2.82%, $P = 0.021$). Correlation tests of ancestry and total somatic mutations also supported the findings (false-discovery rate (FDR) = 0.009 and FDR = 0.032 for SNVs and PGA, respectively; Extended Data Fig. 1d). The top six highest estimates of SV breakpoints per sample were observed among African patients (928–2,284 breakpoints). No overall differences between the ancestries were observed for chromothripsis (range, 52–55%) and chromoplexy (range, 33–38%), whereas tumours from African individuals demonstrated a trend

towards a higher number of interchromosomal chromoplexic chains (1–6 versus 1–2). Moreover, the magnitude of all types of mutations was strongly correlated with one another (Fig. 1b). Thus, the more mutations a prostate tumour has of any given type, the more mutations it is likely to have of all types.

## Candidate oncogenic drivers

Prostate cancer is known to have a long tail of oncogenic drivers[18] across the spectrum of different mutational types[8] (Extended Data Fig. 2). Protein-coding mutations, including those that are probably and possibly damaging, were significantly greater in each African individual (PolyPhen-2, 14 versus 11 mutations in a European individual; $P = 0.022$, two-sample $t$-test; exclusion of hypermutated tumours, $P = 0.039$). We identified 482 coding and 167 non-coding drivers defined by the PCAWG consortium[20] (Extended Data Fig. 3a). A median of two (first quartile to third quartile, 2–4) coding drivers was observed in this study (Supplementary Table 2), with one (0–2) appearing to be specific to prostate cancer[7,8,17,18]. The coding driver genes significantly mutated among 183 patients were *FOXA1*, *PTEN*, *SPOP* and *TP53* (10–25 patients, FDR = $1.34 \times 10^{-21}$–$9.44 \times 10^{-5}$), whereas non-coding driver elements included the *FOXA1* 3′ UTR, *SNORD3B-2* small RNA and a regulatory micro RNA promoter at chromosome 22: 38381983 (FDR = $9.12 \times 10^{-13}$, FDR = $6.16 \times 10^{-9}$ and FDR = 0.070, respectively). Recurrent CNAs of all the patients included 137 gains and 129 losses (GISTIC2, FDR < 0.10; Supplementary Table 3) with some spanning driver genes (Extended Data Fig. 3b), such as *DNAH2* (FDR = $2.18 \times 10^{-7}$), *FAM66C* (FDR = $1.30 \times 10^{-9}$), *FOXP1* (FDR = 0.005), *FXR2* (FDR = $2.18 \times 10^{-7}$), *PTEN* (FDR = $9.61 \times 10^{-13}$), *SHBG* (FDR = $2.18 \times 10^{-7}$) and *TP53* (FDR = $2.18 \times 10^{-7}$).

Moreover, a fraction of somatic SVs (2 breakpoints each; 1,328 breakpoints in total) overlapped with 156 driver genes reported as altered by significantly recurrent breakpoints in the PCAWG study[20], while, using a generalized linear model with adjustable background covariates, we identified an additional 100 genes to be significantly affected by SV breakpoints (FDR = $1.3 \times 10^{-43}$–0.097; Extended Data Fig. 3c and Supplementary Table 4). For more than 20% of tumours, SV breakpoints coexisted with other mutational types within *DNAH2*, *ERG*, *FAM66C*, *FXR2*, *PTEN*, *SHBG* and *TP53*. Using optical genome mapping—an alternative non-sequencing method to examine for chromosomal abnormalities[21]—we validated recurrent breakpoints in HLA regions (*DQA1* and *DQB1* genes), identifying translocations between the 3 Mb HLA complex at chromosome 6 and its corresponding HLA alternative contigs (Extended Data Fig. 3d).

Differences in oncogenic driver alterations between ancestries were observed (Fig. 2a,b). Specifically, tumours from African individuals were more likely to have CNAs and mutations in *SETBP1* (frequency = 0.33, odds ratio (OR) = 0.357, $P = 0.012$), *DDX11L1* (frequency = 0.48, OR = 0.24, $P = 0.0001$), *STK19* (frequency = 0.25, OR = 0.215, $P = 0.004$) and *NCOA2* (frequency = 0.51, OR = 0.172, $P = 3.14 \times 10^{-6}$), along with SVs in *PCAT1* (frequency = 0.13, OR = 0.11, $P = 0.012$). By contrast, SVs for *TMPRSS2* (frequency = 0.38, OR = 3.639, $P = 0.0006$) and *ERG* (frequency = 0.34, OR = 3.159, $P = 0.003$) were more notable among Europeans. Although several DNA-damage repair genes and other genes previously associated with African ancestry were not significantly altered between Africans and Europeans in this study, 10 were solely altered in Africans with most in the coding sequence (frequency = 0.009–0.035). All of these data support the inclusion of a larger number of under-represented populations in clinical enrolment for the benefit of precision oncology studies[22].

## Integrative clustering analysis

Molecular subtyping of tumours is a standard approach in cancer genomics to stratify patients into different degrees of somatic alterations in a homogeneous population, with an implication for clinical

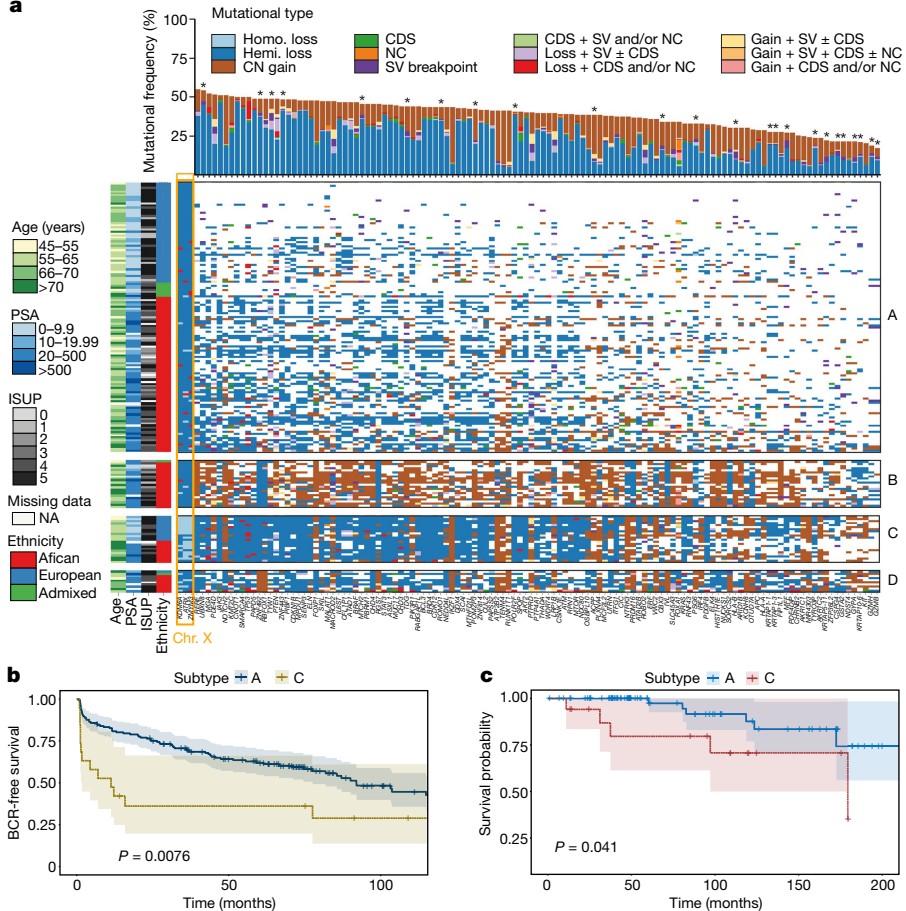

**Fig. 3 | Significance of somatic aberrations across four diverse subtypes.**
**a**, Analysis of the long tail of driver genes using different combinations of mutational types (CDS, coding driver data; NC, non-coding driver data; SV, significantly recurrent breakpoint data; and CN, gene-level CN data), resulting in the identification of 124 preferentially mutated genes among the subtypes. Ordered by mutational frequency, 100 (80.6%) have been reported as significantly recurrent mutations/SV breakpoints in the PCAWG Consortium[20], and 24 (19.4%) are significantly mutated in this study (marked by asterisks). Using iClusterplus, unsupervised hierarchical clustering of all mutational types identified four prostate cancer subtypes (A–D; Fig. 2c), presented for 183 patients (rows) and 124 mutated genes (columns), with each subgroup ordered by ancestry. Ancestrally diverse subtypes A and C are mutationally quiet and are marked by CN loss, respectively. African-specific/predominant subtypes B and D are marked by CN gains and are mutationally noisy, respectively. Three genes on chromosome X, *KDM6A*, *ATRX* and *ZMYM3*, are considered to be significant due to the abundance of homozygous (homo.) loss present in subtype C. Chr., chromosome; hemi., hemizygous; ISUP, International Society of Urologic Pathologists; NA, not applicable. **b**, Kaplan–Meier plot of biochemical relapse (BCR)-free survival proportion of European patients for subtype A (*n* = 161) versus C (*n* = 19). **c**, Kaplan–Meier plot of the cancer survival probability of European patients for subtype A (*n* = 82) versus C (*n* = 17). For **b** and **c**, the probability estimates, 95% confidence intervals and two-sided *P* values (log-rank test) are indicated.

use[9–11]. Identifying five out of the seven TCGA oncogenic driver-defined subtypes in our study[7], European patients were 25% more likely than African patients to be classified (Supplementary Table 5 and Extended Data Fig. 4a–d). Whereas *TMPRSS2-ERG* fusions (predominantly 3 Mb deletions) were significantly elevated in our tumours from European individuals compared with from African individuals (37.7% versus 13.3%; OR = 3.919, *P* = 0.0004), albeit not significantly, African patients were 1.3-fold more likely to present with *SPOP*-coding mutations (MATH and BTB domains).

For further molecular classification, we performed iCluster analysis on all mutational types (small mutations, CNAs and SVs) identifying four subtypes—A to D (Fig. 2c,d and Supplementary Table 6). We found that subtype A is mutationally quiet (1.01 mutations per Mb, 0.50 breakpoints per 10 Mb, 2% PGA); by contrast, subtype D showed the greatest mutational density (1.91 mutations per Mb, 1.08 breakpoints per 10 Mb, 31% PGA) with a mixture of CN gains and losses, whereas subtypes B and C were marked by substantial CN gains or losses, respectively (Fig. 3a). The quiet subtype seems to be common in prostate cancer studies[7,9,23], while the number of pan-cancer consensus drivers[20] increased from subtype A (median, 2 drivers) to B (median, 3 drivers), C (median, 3 drivers) and D (median, 4 drivers).

Using all of the mutational types in the analysis, 124 genes were significantly mutated across the four subtypes (FDR = 3.742 × 10⁻¹³– 0.067; Fig. 3a), occurring in 31 to 183 patients (frequency, 0.17–1). Among them, 100 genes were reported as oncogenic drivers in the PCAWG[20], and *FOXA1* and *SPOP* genes acting as the TCGA subtypes were also replicated in this analysis, while the 24 new mutated genes among the subtypes were predominantly affected by SV breakpoints and CNAs. The median number of mutated genes ranged from 28 (range 3–105) for subtype A to 82, 98 and 93 for subtypes B, C and D, respectively (42–109, 72–112, 49–107). Although different mutational types tended to co-occur within genes and/or patients (Supplementary Table 7), small mutations (coding and non-coding) were noticeably observed in the quiet subtype A, supporting acquisition early in tumorigenesis[24]. Our preferentially mutated genes within tumour subtypes resemble the long tail of prostate cancer drivers[18], with some highly impacting many tumours, but most only affecting a few tumours.

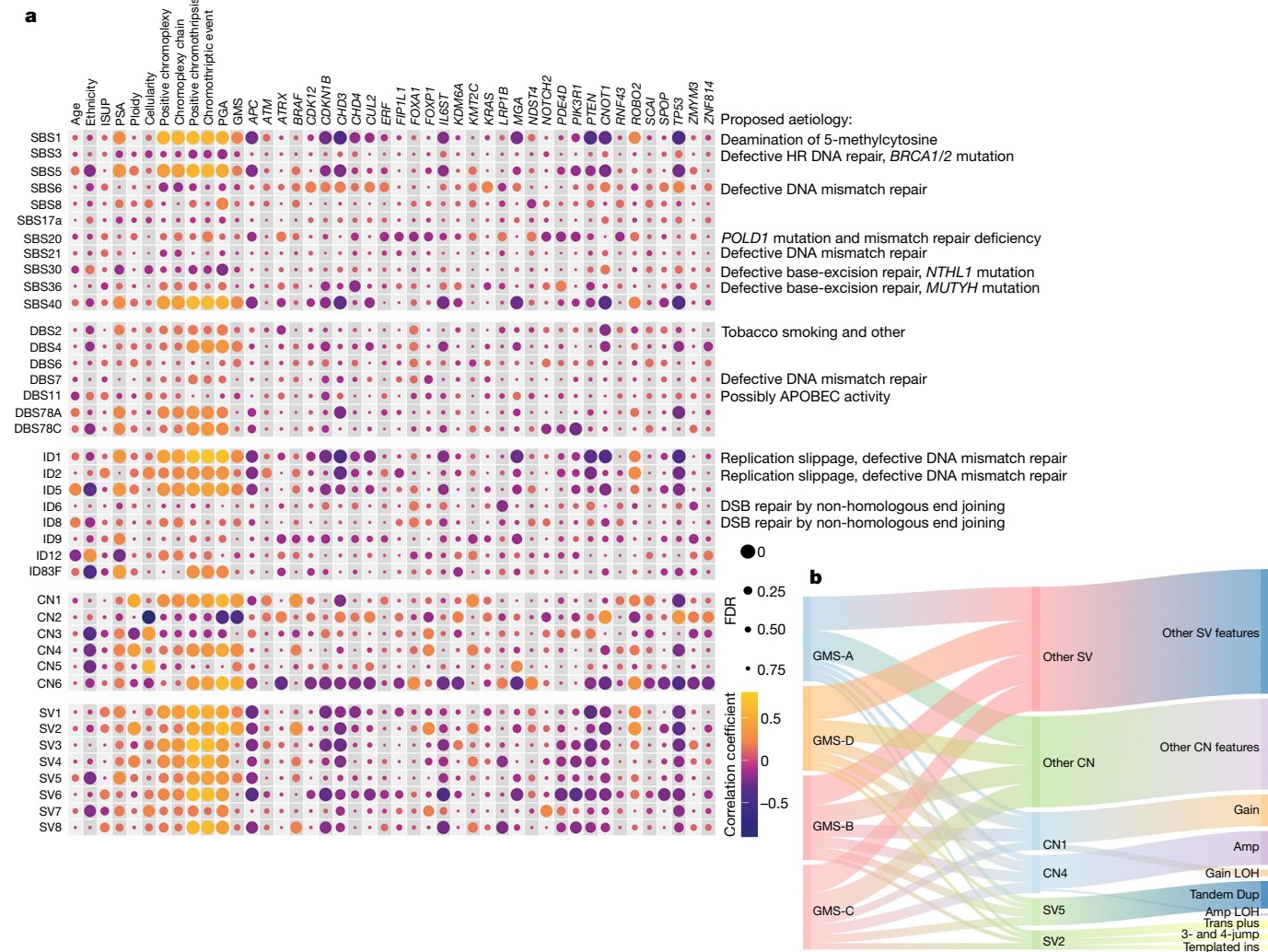

**Fig. 4 | Estimates of genomic aberrations contributed by each mutational signature. a**, Correlation plots of total mutational signatures along with clinical and genomic characteristics. The size of each dot represents the FDR values of Spearman correlation *P* values (two-sided) using Benjamini–Hochberg correction. The colours of each dot represent the correlation coefficient. GMS subtypes are assigned as 1–4 for subtypes A–D, respectively; African, admixed and European are recorded as 1–3, respectively. The correlation of 32 recurrent genes in prostate cancer is shown on the *x* axis. Many small- or large-sized mutational signatures agree with the GMS. HR, homologous recombination; PSA, prostate-specific antigen. **b**, Sankey diagram depicting a

proportion of duplication signatures observed across cancer subtypes. Duplication features, including amplification (Amp), translocation (trans) plus, local *n*-jump, templated insertion (ins), amplification loss of heterozygosity (LOH), gain, tandem duplication and gain LOH (Extended Data Fig. 8a,b) are summed per subtype and equally weighted to 20. Links connecting between nodes (GMS, signatures and features) have widths proportional to the total number of CN or SV features across all patients within each GMS subtype to which they belong. Note that we believe that GMS-B is the identity of the African-specific genomic subtype.

The 124 preferentially mutated genes within our tumour subtypes corresponded to 8 TCGA/ICGC cancer pathways (Supplementary Information and Extended Data Fig. 5). Whereas six showed slightly elevated mutational frequencies in tumours from African individuals, genes affecting epigenetic mechanisms were significantly biased towards European individuals (OR = 5.586, $P = 2.9 \times 10^{-7}$; Extended Data Fig. 6b). Pathway enrichment analysis supported five functional networks of the cancer pathways, with two of them involved in signal transduction and DNA checkpoint processes that five out of the eight pathways interacted with (Extended Data Fig. 6a and Supplementary Table 8).

## Global molecular subtypes

By combining molecular profiling and patient demographics, genetic ancestry and geography, we identified a new prostate cancer taxonomy that we define as GMS (Fig. 2d). Whereas all European patients from Australia (*n* = 53) and Brazil (*n* = 3) were limited to GMS-A and GMS-C,

tumours from African individuals were dispersed across all four subtypes. We found that GMS-B and GMS-D predominate in African individuals, with GMS-B including a single patient of admixed ancestry (92% African) and GMS-D including a single admixed (63% African) and a single European ancestral patient. The latter individual was one of only five Europeans in our study who was born and raised in Africa. Compared with the other patients of European ancestry, this patient showed the highest mutational density across all types. Alternative consensus clustering of individual mutational types mostly recapitulated the subtypes by integrative analysis (Supplementary Table 6). By further including Chinese Asian high-risk prostate cancer data[11] (*n* = 93; Extended Data Fig. 7a), we found that GMS-A is ancestrally and geographically universal, whereas GMS-D remained African specific, with a new African–Asian GMS-E emerging. GMS-B remained African specific and GMS-C remained European–African specific. Although all of the patients were treatment naive at the time of sampling, our European cohort was recruited with extensive follow-up

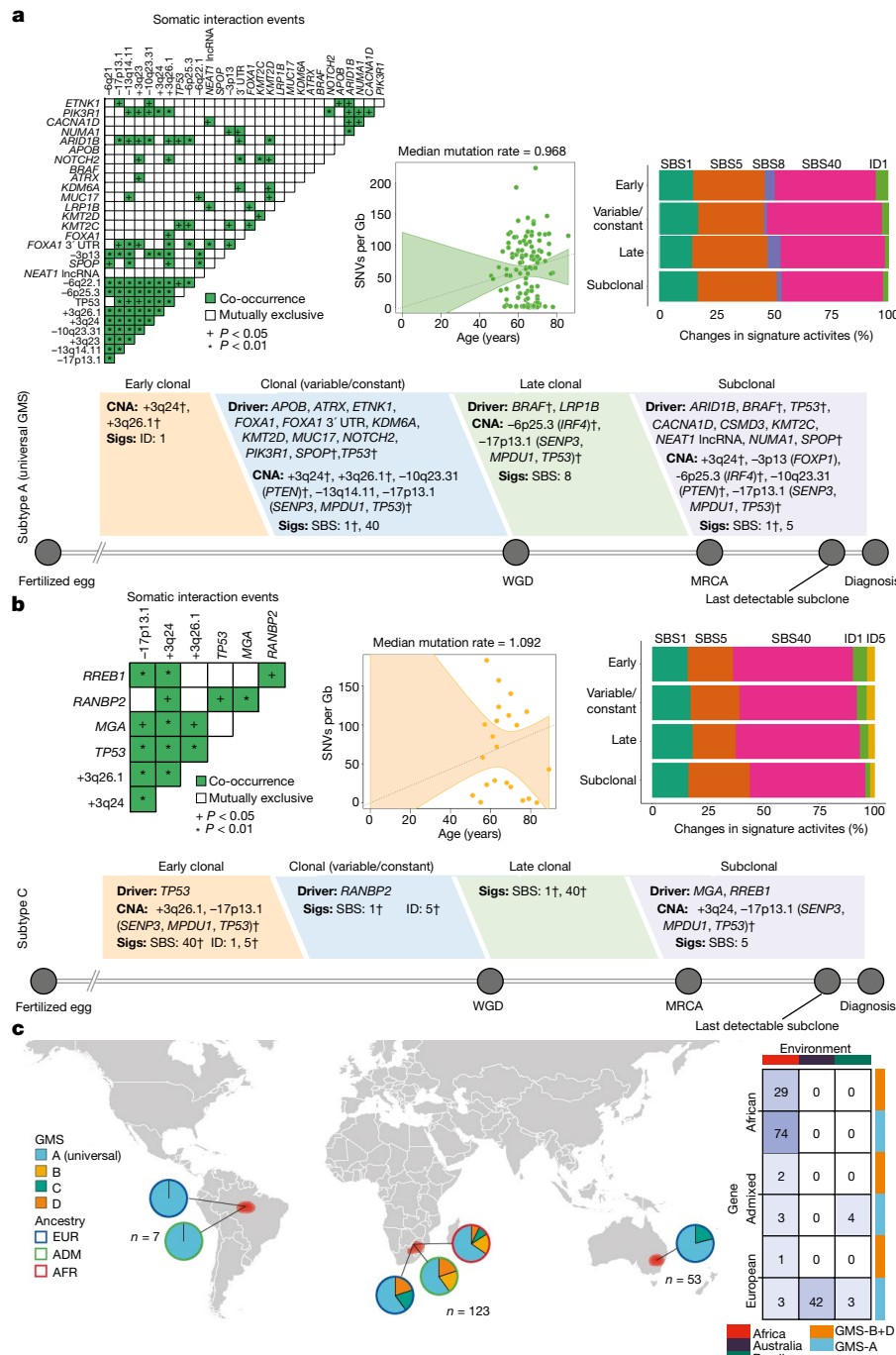

**Fig. 5 | Evolutionary history of globally mutated subtypes. a**, The cancer timeline of the universal subtype (A) begins from the fertilized egg to the age of the patients in a cohort. **b**, The cancer timeline of GMS-C. Estimates for major events, such as whole-genome duplication (WGD) and the emergence of the most-recent common ancestor (MRCA) are used to define the early, variable, late and subclonal stages of tumour evolution approximately in chronological time. When the early and late clonal stages are uncertain, the variable stage is assigned. Driver genes and CNAs are shown in each stage if present in previous studies[8,20] and defined by the MutationTime.R program. Mutational signatures (Sigs) that, on average, change over the course of tumour evolution, or are substantially active, are shown as described in the Supplementary Information. The dagger symbols denote alterations that are found to have different timing. Significant pairwise interaction events between the mutations and CNAs were computed to support cancer timelines. The OR and two-sided *P* value were

calculated using Fisher exact tests. Co-occurrence or mutually exclusive event is considered when OR > 2 or OR < 0.5, respectively. The interaction significance between pairs in GMS-A and GMS-C has *P* values ranging from $2.04 \times 10^{-30}$ to 0.047 and from $1.64 \times 10^{-27}$ to 0.045, respectively. Median mutation rates of CpG-to-TpG burden per Gb are calculated using the age-adjusted branch length of cancer clones and maximally branching subclones. The mutation rate plots in **a** and **b** show the median ± 2 s.e. of fitted data as dashed lines and error bands, respectively. **c**, Schematic of a world map with the distribution of GMS-A–D among ancestrally/globally diverse populations. The gene–environment interaction of GMS is shown on the right. The contingency table of the number of patients with different ancestries (germline variants) stratified by subtypes and associated with certain geography or environmental exposure (two-sided *P* = 0.0005, Fisher exact test with 2,000 bootstraps).

data (median ± s.d., 122.5 ± 44.4 months). Interestingly, biochemical relapse (Fig. 3b) and death-free survival probability (Fig. 3c) explains better clinical outcomes for patients presenting with the universal over the European-African GMS (GMS-A versus GMS-C, log-rank test, $P = 0.008$ and $P = 0.041$, respectively).

Our GMS taxonomy could leverage pan-cancer studies in the following ways. First, a sampling strategy of patients from the PCAWG project was rather homogeneous in each cancer, therefore inhibiting the discovery of globally restricted subtypes[3,13] (Extended Data Fig. 7b). Second, genetic ancestral[25] and geographical data of patients should be included in molecular profiling of cancers. Finally, the inclusion of ethnic disparity in cancer studies would need to properly address genetic admixture in a sampling cohort, with a too low ancestral cut-off appearing to create highly admixed, but similar, ancestry among individuals, therefore discouraging ethnically diverse samples.

## New and known mutational signatures

Approximating the contribution of mutational signatures to individual cancer genomes facilitates the association of the signatures with exogenous or endogenous mutagen exposures that contribute to the development of human cancer[3]. Here we generated a list of CN and SV signatures and their contributions to prostate cancer using non-negative matrix factorization[26] (Extended Data Fig. 8a,b). Combined with a known catalogue of small mutational signatures, including single-base substitutions (SBSs), doublet base substitutions (DBSs) and indels (IDs), we observed not only a substantial variation in the number of mutational features but also over-representation in tumours from African individuals (Extended Data Fig. 8c). Overall, 96 SBS, 78 DBS and 83 ID features examined had significantly higher totals in African individuals (SBSs, 3,399 versus 2,840 in Europeans, $P = 0.014$; DBSs, 42 versus 32, $P = 0.006$; IDs, 374 versus 360, $P = 0.016$, two-sample $t$-tests). We generated six de novo signatures for each small signature type (median cosine similarity, 0.986, 0.856 and 0.976, respectively), corresponding to 12, 7 and 8 global signatures, respectively (median cosine similarity, 0.966, 0.850 and 0.946, respectively; Extended Data Fig. 9), with 26 likely to be of biological origin (SBS47, possible sequencing artefacts). DBSs accounted for about 1% of the prevalence of SBSs. The CN features were also greater in Africans (CN, 3,971 versus 2,721, $P = 1.92 \times 10^{-8}$; SV, 94 versus 88, $P = 0.100$). The SV features defined in a recent pan-cancer study[26] were each mutually exclusive and included simple SVs (split according to size, replication timing and occurrence at fragile sites), templated insertions (split by size), local $n$-jumps and local–distant clusters. The factorization of a sample-by-mutation spectrum matrix identified six CN signatures (CN1–6) and eight SV signatures (SV1–8), as well as their contributions to each tumour.

We found that the full spectrum of mutational signatures (SBSs, DBSs, IDs, CNs and SVs) supports our newly described GMS. Enrichment records of the top signatures in each tumour were significantly associated type by type with the taxonomic subtypes, except for DBSs ($P = 5.1 \times 10^{-7}$–0.017, one-way analysis of variance (ANOVA) or Fisher exact test; Extended Data Fig. 8d). Regardless of the signature type, 13 out of 40 mutational signatures showed either inverse or proportionate correlations with our GMS (FDR = $4.97 \times 10^{-13}$–0.095, Spearman correlation; Fig. 4a). Duplication signatures, including CN1 (tandem duplication), CN4 (whole-genome duplication), SV2 (insertion) and SV5 (large duplication), were biased to the most mutationally noisy subtype (Extended Data Fig. 8a, b), with CN4 and SV5 frequent in Africans (correlation coefficient = −0.24, FDR = 0.005–0.006). Figure 4b shows that the duplication signatures have at least a 1.5× greater proportion of genomic aberrations in GMS-B, GMS-C and GMS-D compared with the universal GMS-A. Furthermore, the African-specific subtype GMS-B consisted of several CN4 and SV5 genomic aberrations composed predominantly of CN amplification (>5 copies and mainly >100 kb in length) and tandem duplication (<5 Mb in size occurred during early to late timing of DNA replication), respectively. Moreover, the mutational density of 30 out of 32 genes that are highly mutated in our GMS and reported in prostate cancer was significantly correlated with different somatic signatures, with most observed in CN2, CN6 and SV6 signatures that were mainly caused by deleted genomes (FDR = $1.61 \times 10^{-7}$–0.082).

## Evolution of GMS

Timeline estimates of individual somatic events reflect evolutionary periods that differ from one patient to another; for example, a cluster of identical alterations derived from clones in one patient presented as subclonal events in another patient (Extended Data Fig. 10a,b). However, they provide in part the order of driver mutations and CNAs present in each sample[24]. The reconstruction of aggregating single-sample ordering of all drivers and CNAs reveals different evolutionary patterns that are unique to each GMS subtype (Fig. 5a,b and Extended Data Fig. 10c). We drew approximate cancer timelines for each GMS subtype portraying the ordering of driver genes, recurrent CNAs and signature activities chronologically interleaved with whole-genome duplication and the emergence of the most recent common ancestor leading up to diagnosis. Basically, significantly co-occurring interactions of the drivers and CNAs are shown (OR = 2.6–97.8, $P = 2.04 \times 10^{-30}$–0.01), supporting their clonal and subclonal ordering states within the reconstructed timelines. SBS and indel signatures that are abundant in each GMS subtype display changes in their mutational spectrum between the clonal and subclonal state, suggesting a difference in mutation rates. The plot of clock-like CpG-to-TpG mutations and patient-age adjustment shows a median mutation rate of as low as 0.968 per year for the universal GMS, but a highest rate of 1.315 per year observed in the African-individual-specific GMS-D. GMS-B and GMS-C have rates of 1.144 and 1.092 per year, respectively. Assessing the relative timing of somatic driver events, TP53 mutations and accompanying 17p loss are of particular interest, occurring early in GMS-C progression and at a later stage in GMS-A. League model relative timing of driver events (Supplementary Information) is consistent with a fraction of probability distribution of the TP53 alterations at the early stage, but most are at an intermediate state of evolution (Extended Data Fig. 10d). This basic knowledge of in vivo tumour development suggests that some tumours could have a shorter latency period before reaching their malignant potential, so known genomic heterogeneity of their primary clones is paramount to pave the way for early detection.

## Discussion

Our study represents one of the largest whole-genome prostate cancer genome resources for sub-Saharan Africa (a summary is provided in Supplementary Table 12). Acknowledging the lack of information on clinical staging for the South African patients (recruited at diagnosis), we describe a prostate cancer molecular taxonomy, identifying ancestrally distinctive GMS. Compared to previous taxonomy using significantly mutated genes in prostate cancer[7,18], we found that GMS subtypes compliment known subtypes such as SPOP and FOXA1 mutations, in contrast to under-represented subtypes in this study, including gene fusions (Extended Data Fig. 4a). We also found that GMS subtypes correlate with mutational signatures reported in the known catalogue of somatic mutations in cancer, in which each tumour is represented by different degrees of exogenous and endogenous mutagen exposures[3]. Our study used the analysis of evolution across 38 cancer types by the PCAWG consortium[24], recognizing that each GMS subtype represents a unique evolutionary history with drivers and mutational signatures varied between cancer stages and linking somatic evolution to a patient's demographics. Thus, some represent rare or geographically restricted signatures that have not been observed in pan-cancer studies[3,13].

We considered two extreme cases, universal GMS-A versus African-specific GMS-B and GMS-D, that would have been influenced by two

different mutational processes for conceptual simplicity (Fig. 5c). One factor is predisposing genetics[27–29] contributing to endogenous mutational processes, especially those with significant germline–somatic interactions, such as the *TMPRSS2-ERG* fusion that is less frequently observed in men of African and Asian ancestry[11,30], germline *BRCA2* mutations and the somatic *SPOP* driver co-occurred with their respective counterparts[31,32]. Another factor is modifiable environmental attributes that are specific to certain circumstances or geographical regions that, to date, have not been observed in prostate cancer. They act as mutagenic forces leading to the positive selection of point mutations throughout life in healthy tissues[33,34] and cancers[35], forming fluid boundaries between normal ageing and cancer tissues. According to Ottman[36], the above-mentioned model of gene–environment interaction is observed when there is a different effect of a genotype on disease in individuals with different environmental exposures or, alternatively, a different effect of an environmental exposure on disease in individuals with different genotypes. Other GMS subtypes would be a combination of the two processes, warranting a need for larger populations capturing ancestral versus ethnic and geographical diversity. As such, the study directly accounts for the large spatiogenomic heterogeneity of prostate cancer and its associated evolutionary history in understanding the disease aetiology.

Our study suggests that larger genomic datasets of geo-ethnically diverse and ancestrally defined populations in a unified analysis will continue to identify rare and geographically restricted subtypes in prostate cancer and potentially other cancers. We demonstrate that ancestral and geographical attributes of patients could facilitate those studies on cancer population genomics, an alternative to cancer personalized genomics, for a better scientific understanding of nature versus nurture.

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

## Methods

### Patient cohorts and WGS

Our study included 183 treatment-naive patients with prostate cancer who were recruited under informed consent and appropriate ethics approval (Supplementary Information 2) from Australia ($n$ = 53), Brazil ($n$ = 7) and South Africa ($n$ = 123). While matched for pathological grading, as previously reported, prostate-specific antigen levels are notably elevated within our African patients[16] and we cannot exclude on the basis of potential metastasis (as data on metastases in this cohort are unavailable). DNA extracted from fresh tissue and matched blood underwent 2 × 150 bp sequencing on the Illumina NovaSeq instrument (Kinghorn Centre for Clinical Genomics, Garvan Institute of Medical Research).

### WGS processing and variant calling

Each lane of raw sequencing reads was aligned against human reference hg38 + alternative contigs using bwa (v.0.7.15)[37]. Lane-level BAM files from the same library were merged, and duplicate reads were marked. The Genome Analysis Toolkit (GATK, v.4.1.2.0) was used for base quality recalibration[38]. Contaminated and duplicate samples ($n$ = 8) were removed. We implemented three main pipelines for the discovery of germline and somatic variants, with the latter including small (SNV and indel) to large genomic variation (CNAs and SVs). The complete pipelines and tools used are available from the Sydney Informatics Hub (SIH), Core Research Facilities, University of Sydney (see the 'Code availability' section). Scalable bioinformatic workflows are described in Supplementary Information 4.

Genetic ancestry was estimated using fastSTRUCTURE (v.1.0)[39], Bayesian inference for the best approximation of marginal likelihood of a very large variant dataset. Reference panels for African and European ancestry compared in this study were retrieved from previous whole-genome databases[19].

### Analysis of chromothripsis and chromoplexy

Clustered genomic rearrangements of prostate tumours were identified using ShatterSeek (v.0.4)[40] and ChainFinder (v.1.0.1)[41]. Our somatic SV and somatic CNA call sets were prepared and co-analysed using custom scripts (see the 'Code availability' section; Supplementary Information 6).

### Analysis of mutational recurrence

We used three approaches to detect recurrently mutated genes or regions based on three mutational types, including small mutations, SVs and CNAs (Supplementary Information 7). In brief, small mutations were tested within a given genomic element as being significantly more mutated than the adjacent background sequences. The genomic elements retrieved from syn5259886, the PCAWG Consortium[20], were a group of coding sequences and ten groups of non-coding regions. SV breakpoints were tested in a given gene for their statistical enrichment using gamma–Poisson regression and corrected by genomic covariates[12]. Focal and arm-level recurrent CNAs were examined using GISTIC (v.2.0.23)[42]. Known driver mutations in coding and non-coding regions published in PCAWG[20,43,44] were also recorded in our 183 tumours, and those specific to prostate cancer genes were also included[7,8,12,17,18].

### Integrative analysis of prostate cancer subtypes

Integrative clustering of three genomic data types for 183 patients was performed using iClusterplus[11,45] in R, with the following inputs: (1) driver genes and elements; (2) somatic CN segments; and (3) significantly recurrent SV breakpoints. We ran iClusterPlus.tune with clusters ranging from 1 to 9. We also performed unsupervised consensus clustering on each of the three data types individually. Association analysis of genomic alteration with different iCluster subtypes was performed in detail (Supplementary Information 8). Differences in driver mutations, recurrent breakpoints and somatic CNAs across different iCluster subtypes were reported.

### Comparison of iCluster with Asian and pan-cancer data

To compare molecular subtypes between extant human populations, the Chinese Prostate Cancer Genome and Epigenome Atlas (CPGEA, PRJCA001124)[11] was merged and processed with our integrative clustering analysis across the three data types described above, with some modifications. Moreover, we leveraged the PCAWG consortium data[13] to define molecular subtypes across different ethnic groups in other cancer types using published data of somatic mutations, SV and GISTIC results by gene. Four cancer types consisting of breast, liver, ovarian and pancreatic cancers were considered due to existing primary ancestries of African, Asian and European with at least 70% contribution. Full details are provided in Supplementary Information 8.4.

PCAWG[13] participants with prostate cancer were retrieved to compare with Australian data with clinical follow-up. Only those of European ancestry greater than 90% ($n$ = 139) were analysed for the three genomic data types of iCluster subtyping, as well as individual consensus clustering. Clustering results identical to the larger cohort size mentioned above were chosen for association analyses. Differences in the biochemical relapse and lethal prostate cancer of the participants across the subtypes were assessed using the Kaplan–Meier plot followed by a log-rank test for significance.

### Analysis of mutational signatures

Mutational signatures (SBSs, DBSs and indels), as defined by the PCAWG Mutational Signatures Working Group[3], were fit to individual tumours with observed signature activities using SigProfiler[46]. Non-negative matrix factorization was implemented to detect de novo and global signature profiles among 183 patients and their contributions. New mutational genome rearrangement signatures (CN and SV) were also performed using non-negative matrix factorization, with 45 CN and 44 SV features examined across 183 tumours. We followed the PCAWG working classification and annotation scheme for genomic rearrangement[26]. Two SV callers were used to obtain exact breakpoint coordinates. Replication timing scores influencing on SV detection were set at >75, 20–75 and <20 for early, mid, and late timing, respectively[47]. Full details of analysis steps, parameters and relevant statistical tests are provided in Supplementary Information 9.

### Reconstruction of cancer timelines

Timing of CN gains and driver mutations (SNVs and indels) into four epochs of cancer evolution (early clonal, unspecified clonal, late clonal and subclonal) was conducted using MutationTimeR[24]. CN gains including 2 + 0, 2 + 1 and 2 + 2 (1 + 1 for a diploid genome) were considered for a clearer boundary between epochs instead of solely information of variant allele frequency. Confidence intervals ($t_{lo} - t_{up}$) for timing estimates were calculated with 200 bootstraps. Mutation rates for each subtype were calculated according to ref. [24] such that CpG-to-TpG mutations were counted for the analysis because they were attributed to spontaneous deamination of 5-methyl-cytosine to thymine at CpG dinucleotides, therefore acting as a molecular clock.

League model relative ordering was performed to aggregate across all study samples to calculate the overall ranking of driver mutations and recurrent CNAs. The information for the ranking was derived from the timing of each driver mutation and that of clonal and subclonal CN segments, as described above. A full description is provided in Supplementary Information 10.

### Reporting summary

Further information on research design is available in the Nature Research Reporting Summary linked to this article.

## Data availability

DNA-sequencing data have been deposited at the European Genome-Phenome Archive (EGA) under overarching accession EGAS00001006425 and including the Southern African Prostate Cancer Study (SAPCS) Dataset (EGAD00001009067 and Garvan/St Vincent's Prostate Cancer Database EGAD00001009066). Academic researchers meeting the data-access policy criteria may apply for data access through the respective data access committees. CPGEA data are available through http://www.cpgea.com. PCAWG data are available at ICGC Data Portal (https://dcc.icgc.org/releases/PCAWG).

## Code availability

The core computational pipelines used in this study for read alignment, quality control and variant calling are available at GitHub (https://github.com/Sydney-Informatics-Hub/Bioinformatics). Analysis code for chromothripsis and chromoplexy is also available at GitHub (https://github.com/tgong1/Code_HRPCa).

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

**Acknowledgements** The work presented was supported by the National Health and Medical Research Council (NHMRC) of Australia through a Project Grant (APP1165762, to V.M.H.), NHMRC Ideas Grants (APP2001098, to V.M.H. and M.S.R.B.; APP2010551, to V.M.H.), University of Sydney Bridging Grant (G199756, to V.M.H.), and through the US Department of Defense (DoD) Prostate Cancer Research Program (PCRP) Idea Development Award TARGET Africa (PC200390, to W.J., S.M.P., D.C.W., S.B.A.M., M.S.R.B. and V.M.H.). We acknowledge the use of the National Computational Infrastructure (NCI), which is supported by the Australian Government, and accessed through the National Computational Merit Allocation Scheme (V.M.H., E.K.F.C. and W.J.), the Intersect Computational Merit Allocation Scheme (V.M.H.), Intersect Australia Limited and the Sydney Informatics Hub, Core Research Facility, and we acknowledge the staff at the Garvan Institute of Medical Research's Kinghorn Centre for Clinical Genomics (KCCG) core facility for genome sequencing. Recruitment, sampling and processing for the Southern African Prostate Cancer Study (SAPCS), as required for the purpose of this study, was supported by the Cancer Association of South Africa (CANSA; M.S.R.B. and V.M.H.). V.M.H. is supported by the Petre Foundation through the University of Sydney Foundation; A.-M.H. and W.J. by a Cancer Institute of New South Wales (CINSW) Program Grant (TPG172146 to L.G.H., J.G.K., P.D.S. and V.M.H.), with additional support to W.J. provided by the Prostate Cancer Research Alliance Australian Government and Movember Foundation Collaboration PRECEPT (prostate cancer prognosis and treatment study, led by N. Corcoran). We thank the patients and their families who contributed to this study; without their contribution, this research would not be possible; the many clinical staff across the SAPCS (South Africa), the St Vincent's Hospital Sydney (Australia) and Endocrine and Tumor Molecular Biology Laboratory (Brazil) for their contributions, who over many years have recruited patients and provided samples to these critical bioresources, with special recognition of P. Venter, R. L. Monare and S. van Zyl for their contributions as inaugural members of the SAPCS.

**Author contributions** V.M.H. designed the experiments and supervised the project. W.J. led the bioinformatic and statistical analyses, and both W.J. and V.M.H. performed data interpretation. S.M.P., R.J.L., A.-M.H. and D.G.P. prepared the samples and managed phenotypic data. M.L. and J.G.K. performed pathological grading, and R.C., L.G.H., I.S.B., S.B.A.M., P.D.S. and M.S.R.B. managed patient recruitments and consents, as well as clinical interpretation. V.M.H., S.B.A.M. and M.S.R.B. codirect the Southern African Prostate Cancer Study (SAPCS). W.J., J.J., T.G., C.W., T.C. and R.S. developed the pipelines, and performed the efficient and scalable high-performance computational variant calling, with critical advice provided by E.K.F.C. and V.M.H.; W.J., J.J. and T.G. performed complex variant annotation, and R.J.L. generated the optical genome mapping data. W.J. performed mutational signature and tumour evolution analysis, with critical advice provided by D.C.W.; W.J. and V.M.H. wrote the manuscript. W.J. generated the figures, and all of the authors contributed to the final editing and approval of the manuscript.

**Competing interests** The authors declare no competing interests.

**Additional information**
**Correspondence and requests for materials** should be addressed to Vanessa M. Hayes.

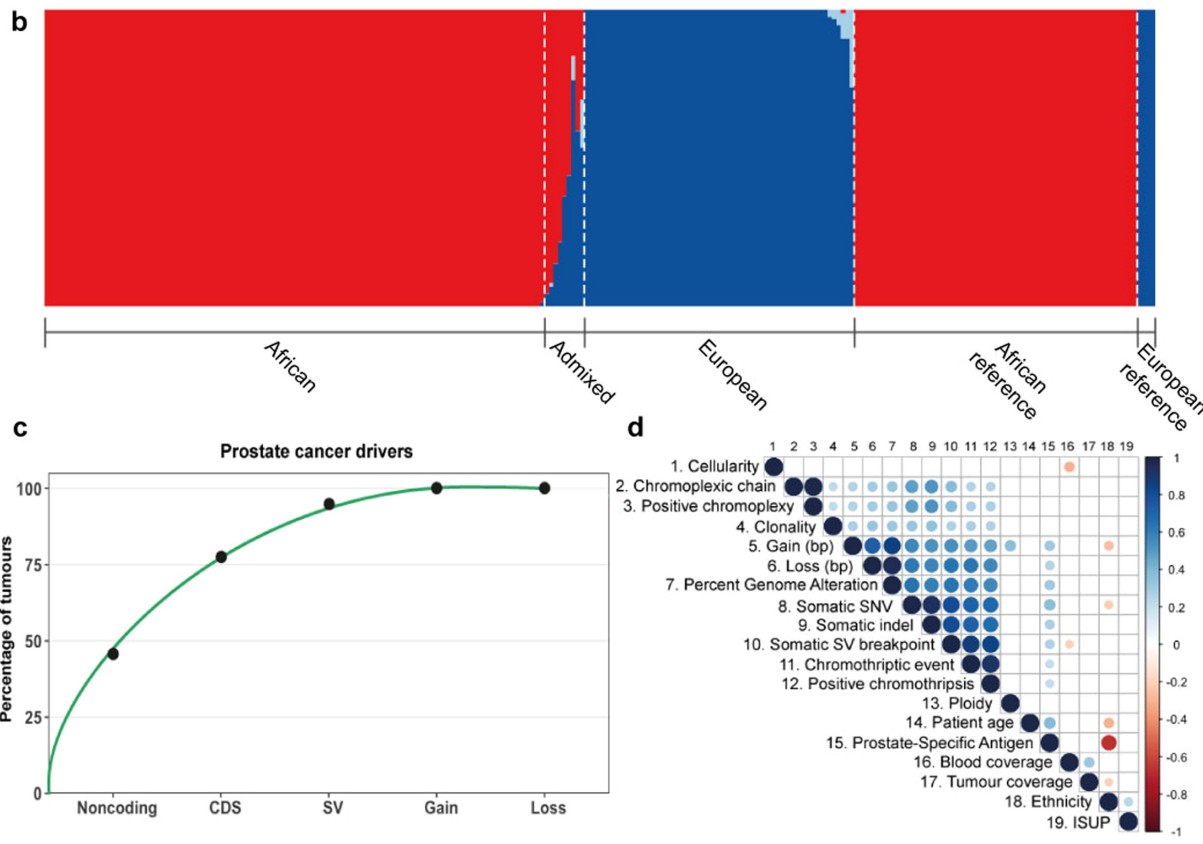

**a**

| | African ancestry | European ancestry | Admixed ancestry |
|---|---|---|---|
| **Country of origin** | **113 patients**[***] | **61 patients**[***] | **9 patients**[***] |
| South Africa | 113 | 5 | 5 |
| Australia | 0 | 53 | 0 |
| Brazil | 0 | 3 | 4 |
| **Clinical age (median, Q1-Q3)** | **68 years (63-71)**[**] | **63 years (58-67)**[**] | 64 years (59-71) |
| 45 - 60 years | 21 | 22 | 3 |
| 61 - 75 years | 77 | 38 | 5 |
| ≥ 76 years | 14 | 1 | 1 |
| **Pre-operative PSA (median, Q1-Q3)** | **52.2 ng/ml (24-133)**[***,*] | **8.2 ng/ml (6-12)**[***] | **17.4 ng/ml (10-79)**[*] |
| < 10 ng/ml | 9 | 40 | 2 |
| 10 - 20 ng/ml | 12 | 17 | 3 |
| ≥ 20 ng/ml | 86 | 4 | 4 |
| **ISUP grade group (median, Q1-Q3)** | **GG 4 (GG 3-5)**[***] | **GG 5 (GG 4-5)**[***] | GG 4 (GG 4-5) |
| GG 0 - 2 | 19 | 8 | 2 |
| GG 3 | 10 | 1 | 0 |
| GG 4 | 38 | 11 | 3 |
| GG 5 | 38 | 41 | 4 |
| **WGS tumour purity (median, Q1-Q3)** | **48 % (42-52 %)**[**] | **45 % (39-56 %)**[**] | 44 % (42-45 %) |
| 10 - 40 % | 24 | 23 | 1 |
| 41 - 60 % | 80 | 26 | 8 |
| 61 - 90 % | 9 | 12 | 0 |

**b**

African | Admixed | European | African reference | European reference

**c**

Prostate cancer drivers

Percentage of tumours

Noncoding | CDS | SV | Gain | Loss

**d**

1. Cellularity
2. Chromoplexic chain
3. Positive chromoplexy
4. Clonality
5. Gain (bp)
6. Loss (bp)
7. Percent Genome Alteration
8. Somatic SNV
9. Somatic indel
10. Somatic SV breakpoint
11. Chromothriptic event
12. Positive chromothripsis
13. Ploidy
14. Patient age
15. Prostate-Specific Antigen
16. Blood coverage
17. Tumour coverage
18. Ethnicity
19. ISUP

**Extended Data Fig. 1 | Clinical cohorts and statistical metrics. a**, Clinical and pathological patient. characterization. Pairwise comparisons using contingency tables and Fisher's Exact test between African ancestry and Admixed/European ancestry are highlighted in bold with two-sided $P$-value <0.05 (*), <0.01 (**), or <0.001 (***). Summary statistics, including the median, first and third quartiles (Q1-Q3), are also present. **b**, STRUCTURE analysis of bi-allelic germline variants with the logistic prior model. Model components used to explain structure in the plot are K = 5. All spectrum of African contributions are summed and assigned as African ancestry. **c**, Saturation curve for all driver types across 183 patients. Recurrent copy number gains and losses were measured using GISTIC v2 (Supplementary Methods). CDS, coding sequence; SV, structural variation. **d**, Spearman's correlation between different variables measured in this cohort. Dot sizes represent the magnitude of correlation, with significant $P$-values (two-sided) <0.01.

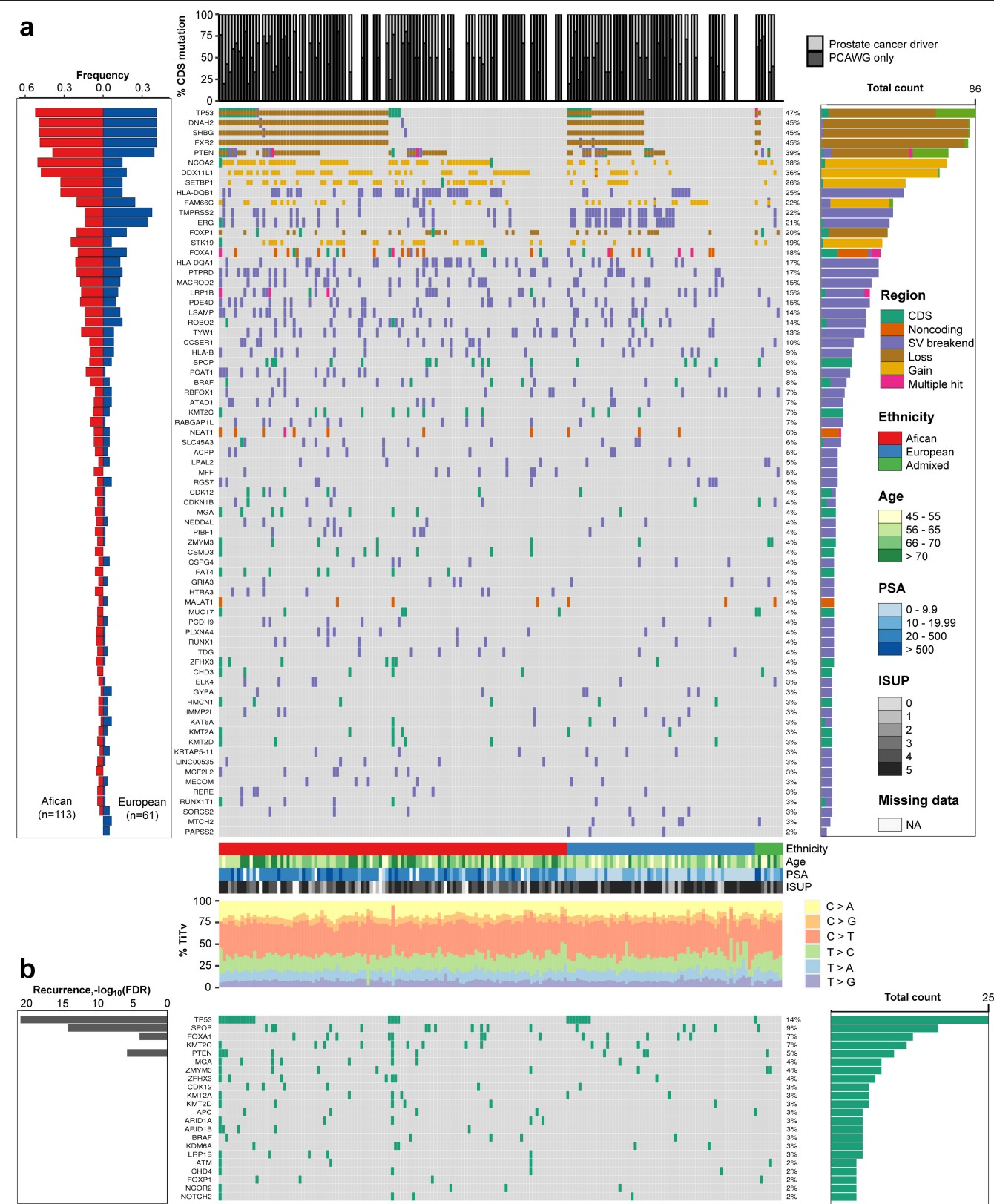

**Extended Data Fig. 2 | Somatic driver mutations in 183 prostate cancer patients of different ancestries.** The covariates on the right show the total number of altered samples for different mutational types. **a**, Search of the top 300 driver genes altered in primary prostate tumours among 183 specimens. Only driver genes discovered in PCAWG and this study, present in more than six patients or significantly different between Africans and Europeans are chosen for plotting. The top barplot shows the distribution of the number of prostate cancer drivers and/or that of PCAWG. The heatmap shows drivers found in this study (rows) for each patient (columns). Heatmaps are coloured by mutational type. The dual barplot on the left depicts gene-level comparisons of mutational recurrence directly between Africans and Europeans. Bottom covariates show the clinical features of patients. The percentage of transition/transversion mutations across 183 patients shows 1,364,210 small somatic mutations across chromosomes 1-Y. **b**, The bottom heatmap shows the top 22 of previously reported coding driver genes in prostate cancer observed in this study[7,8,17,18]. The left barplot shows statistical support of recurrence analysis for our study.

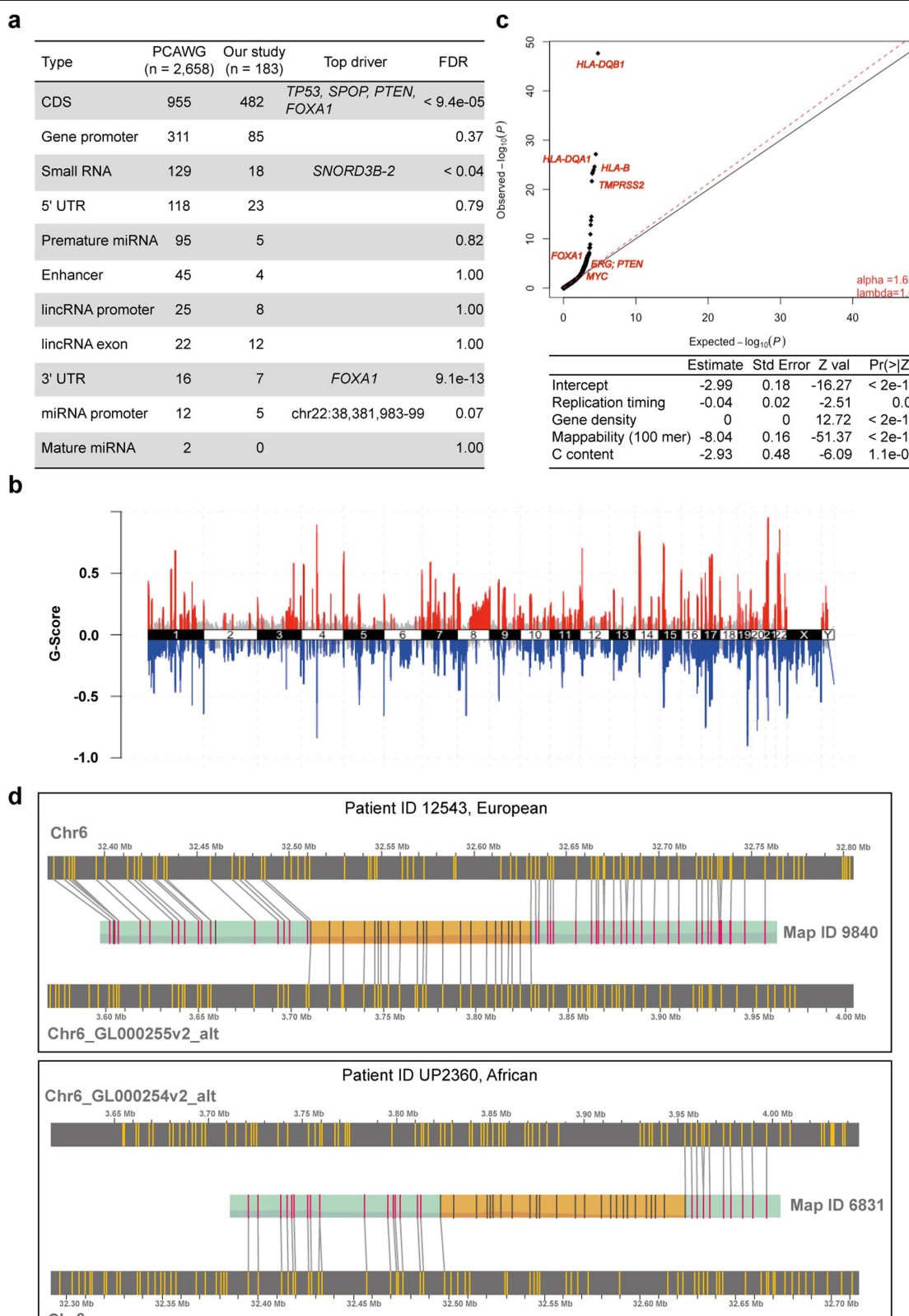

**a**

| Type | PCAWG (n = 2,658) | Our study (n = 183) | Top driver | FDR |
|---|---|---|---|---|
| CDS | 955 | 482 | *TP53, SPOP, PTEN, FOXA1* | < 9.4e-05 |
| Gene promoter | 311 | 85 | | 0.37 |
| Small RNA | 129 | 18 | *SNORD3B-2* | < 0.04 |
| 5' UTR | 118 | 23 | | 0.79 |
| Premature miRNA | 95 | 5 | | 0.82 |
| Enhancer | 45 | 4 | | 1.00 |
| lincRNA promoter | 25 | 8 | | 1.00 |
| lincRNA exon | 22 | 12 | | 1.00 |
| 3' UTR | 16 | 7 | *FOXA1* | 9.1e-13 |
| miRNA promoter | 12 | 5 | chr22:38,381,983-99 | 0.07 |
| Mature miRNA | 2 | 0 | | 1.00 |

**c**

| | Estimate | Std Error | Z val | Pr(>|Z|) |
|---|---|---|---|---|
| Intercept | -2.99 | 0.18 | -16.27 | < 2e-16** |
| Replication timing | -0.04 | 0.02 | -2.51 | 0.01* |
| Gene density | 0 | 0 | 12.72 | < 2e-16** |
| Mappability (100 mer) | -8.04 | 0.16 | -51.37 | < 2e-16** |
| C content | -2.93 | 0.48 | -6.09 | 1.1e-09** |

**Extended Data Fig. 3** | See next page for caption.

**Extended Data Fig. 3 | Discovery of prostate cancer drivers. a**, The number and types of PCAWG driver genes and elements studied in our cohort. **b**, Recurrent copy number alterations among 183 prostate tumours identified with a 99% confidence level using GISTIC v2 (Supplementary Methods). The figure shows GISTIC peaks of significant regions of recurrent amplification (red) or deletion (blue) supported by FDR < 0.01. **c**, Genome-wide scan for significantly recurrent breakpoints in our study. The quantile-quantile plot shows two-sided *P*-values for mutational densities across 183 prostate cancer patients. Multiple hypothesis corrections using the false discovery rate (FDR; Benjamini–Hochberg method) are shown in Supplementary Table 4. Generalized linear modelling (GLM) of somatic mutation densities along the genome with significant background mutational processes adjusted in the model is also shown. **d**, Bionano Genomics optical genome mapping at the HLA complex. Examples of HLA translocations from a European patient (ID 12543) and an African patient (ID UP2360) studied in this cohort are characterized by pairs of optical maps, each carrying a fusion junction with flanking fragments aligning to one side of the two reference breakpoints. Using the recurrent HLA breakpoints identified in this study, the genome map of the African specimen is found to have a low-end fusion function matched with chromosome 6 through a manual inspection of unfiltered consensus maps using Bionano Access v1.5.2. Note that the HLA alternate contig fused in the European tumour is different from one suggested by short-read sequencing (chr6_GL000252v2_alt). The reference genome map is an *in silico* digest of the human reference hg38 with the DLE-1 enzyme. Genome map sizes are indicated on the horizontal axis, in megabase (Mb) units. Matching fluorescent labels between sample and reference genome map are connected by grey lines.

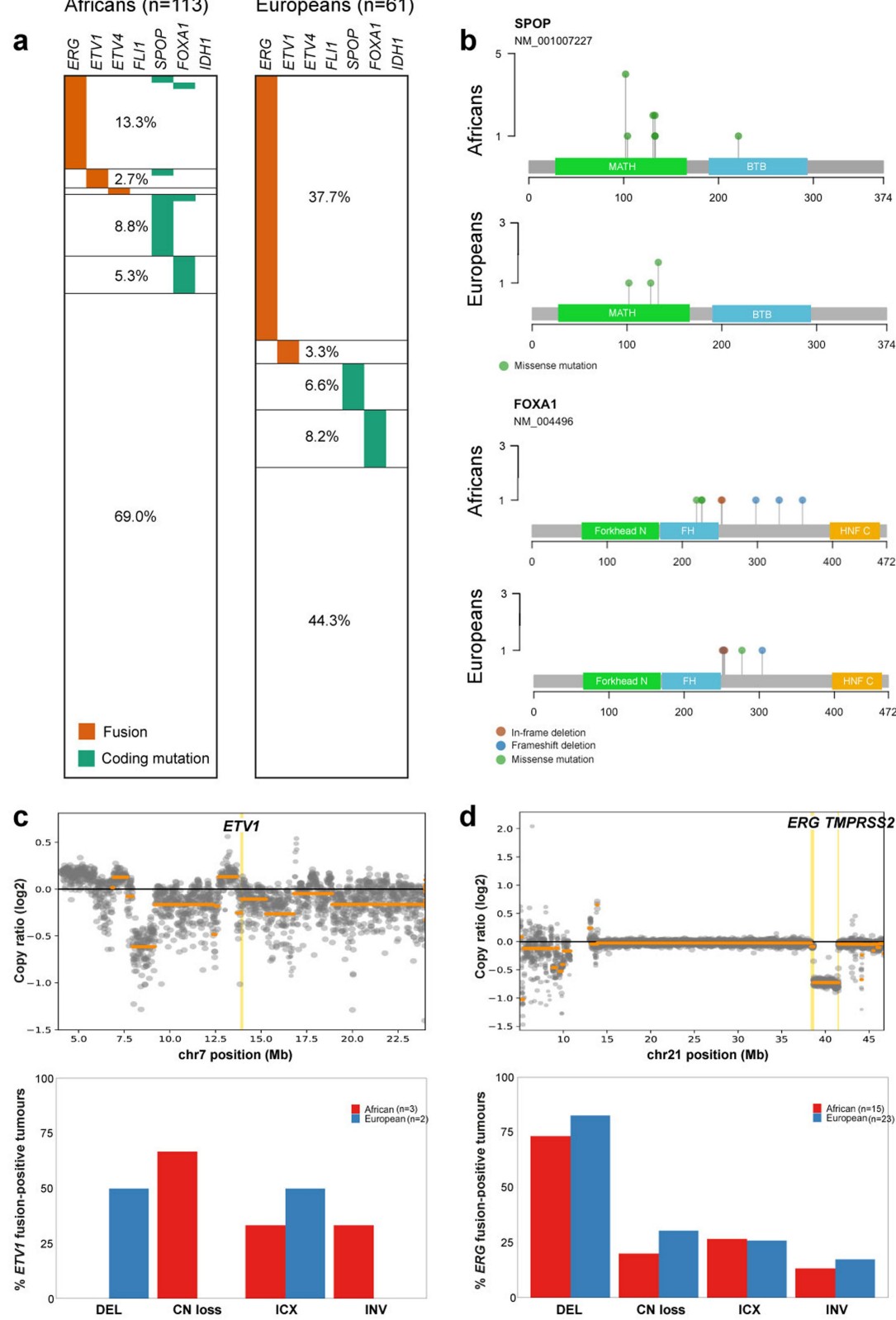

**Extended Data Fig. 4 | TCGA molecular taxonomy. a**, Seven important oncogenic drivers identified by TCGA within our African and European patients. **b**, Coding mutations observed within *SPOP* and *FOXA1* genes. Rarely, a mutation at the BTB domain of *SPOP* gene is shown (R221C in an African patient, KAL0072). FH, forkhead. **c**, *ETV1* fusions within positive patients caused by copy number (CN) losses and/or structural variants (DEL, deletion; ICX, interchromosomal translocation; and INV, unbalanced or balanced inversion). CN changes in chromosome 7 show the *ETV1* loss with $\log_2$ CN ratio less than −0.2. **d**, *ERG* fusions caused by CN losses and/or structural variants.

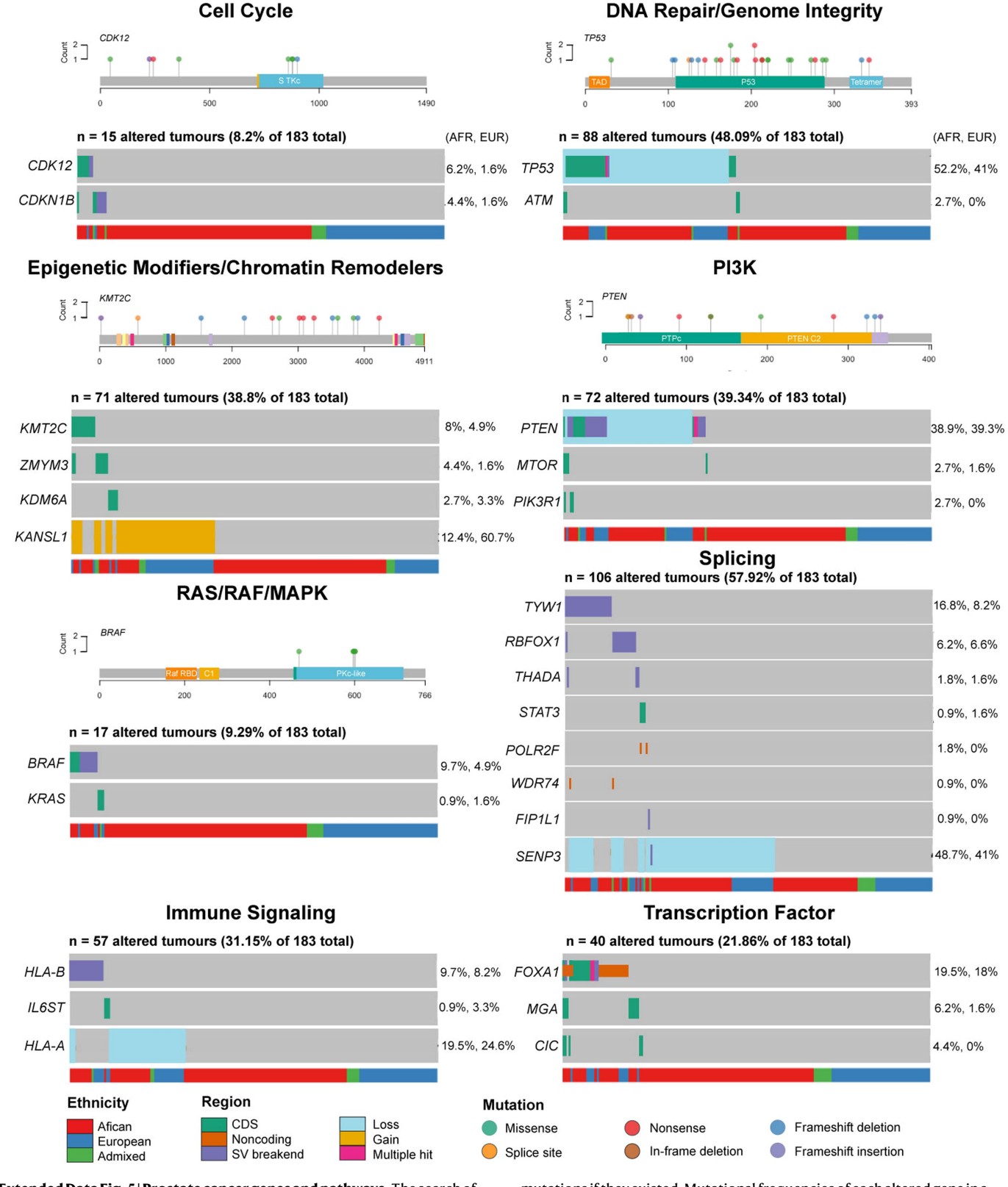

**Extended Data Fig. 5 | Prostate cancer genes and pathways.** The search of our 124 preferentially mutated genes across tumour subtypes is carried out using the TCGA and ICGC cancer databases. The top affected genes for each pathway are present with lollipop plots to show their hotspots of simple coding mutations if they existed. Mutational frequencies of each altered gene in a pathway are separately measured between Africans (n = 113) and Europeans (n = 61) and shown on the right as a percentage in order (AFR, EUR).

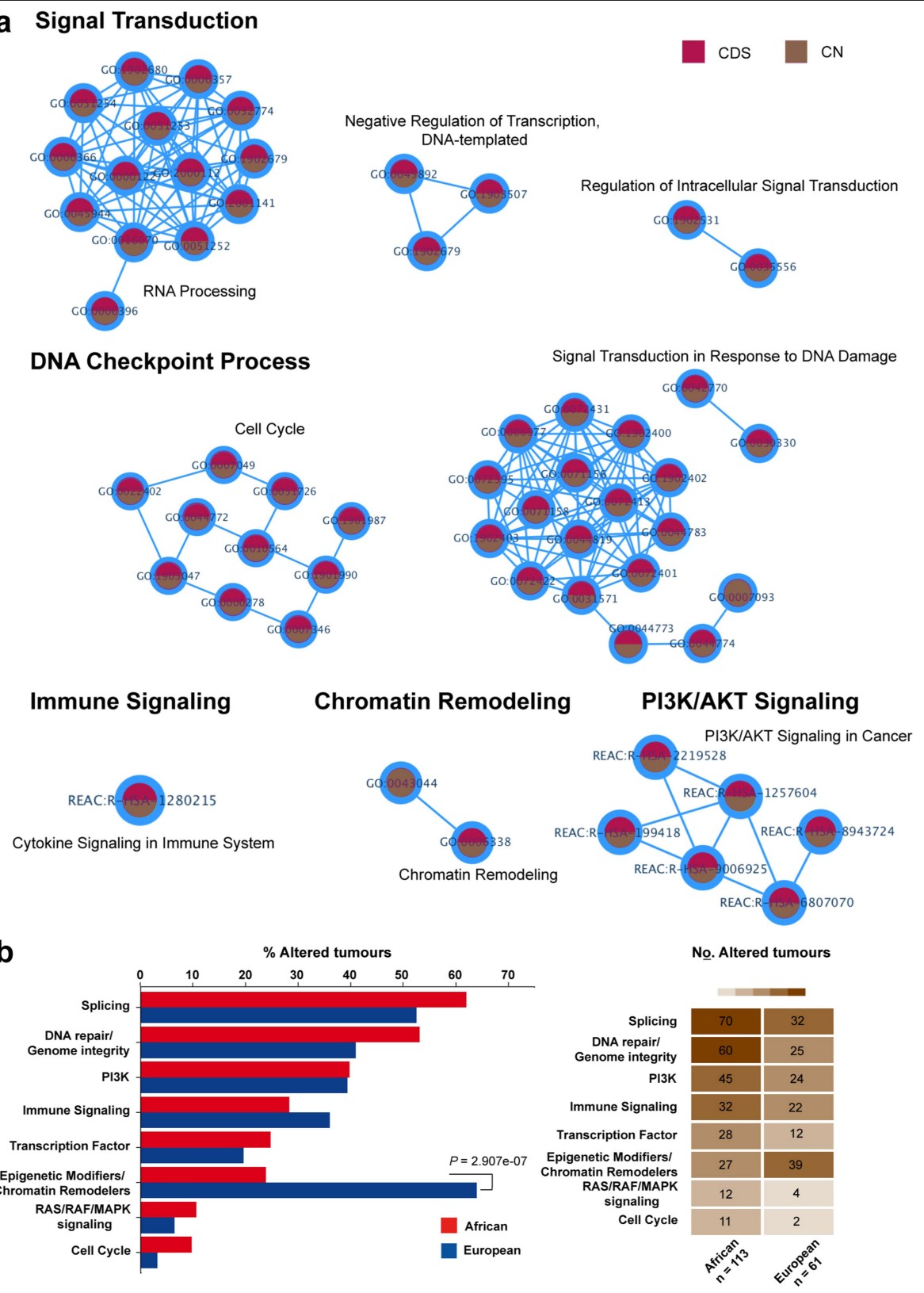

**Extended Data Fig. 6 | Major biological pathways and networks of prostate cancer. a**, Networks of functional interactions between driver genes are shown for each cancer pathway. Nodes represent Gene Ontology biological processes and Reactome pathways and edges show functional interactions. **b**, Pathway alteration frequencies between African and European. A sample was considered altered in a given pathway if at least a single gene in the pathway had a genomic alteration (see Extended Data Fig. 5). *P*-values indicate the level of significance (two-sided Fisher's exact test).

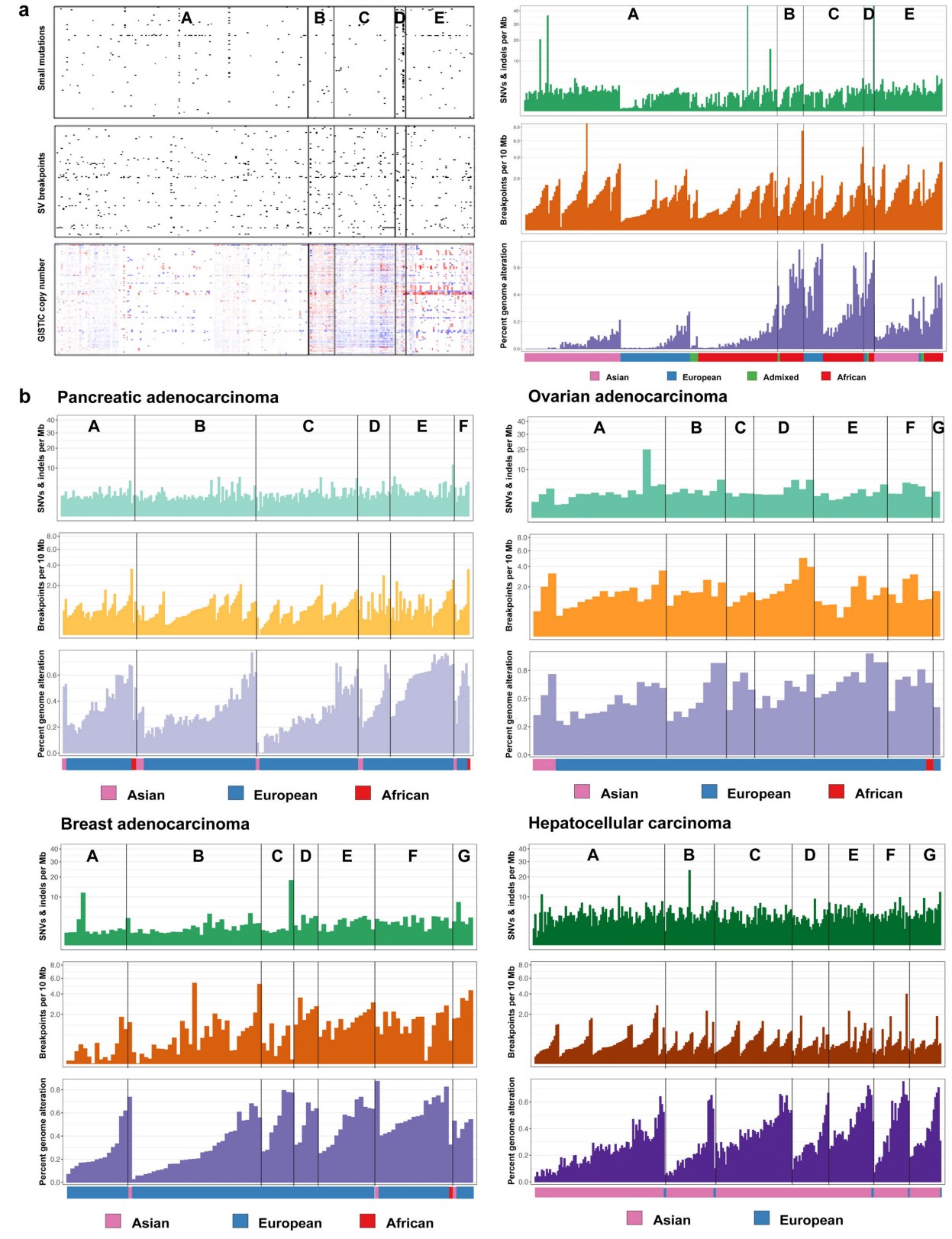

**Extended Data Fig. 7 | Molecular subtypes in prostate cancer and pan-cancers. a**, Unsupervised hierarchical clustering of primary prostate tumours across three major ancestral groups was performed using total somatic mutations present within WGS normalized data. Admixed individuals were also tested in prostate cancer subtypes to which they belonged. **b**, Molecular subtyping of total somatic mutations within pan-cancer studies,

namely pancreatic, ovarian, breast and liver cancers. Raw data of small somatic mutations, structural variants and copy number alterations acquired per cancer were retrieved from the PCAWG[13]. For each subtype, patients are ordered based on their ancestry. Ancestral groups are assigned using a cut-off of ancestral contribution greater than 70%; otherwise, considered as Admixed.

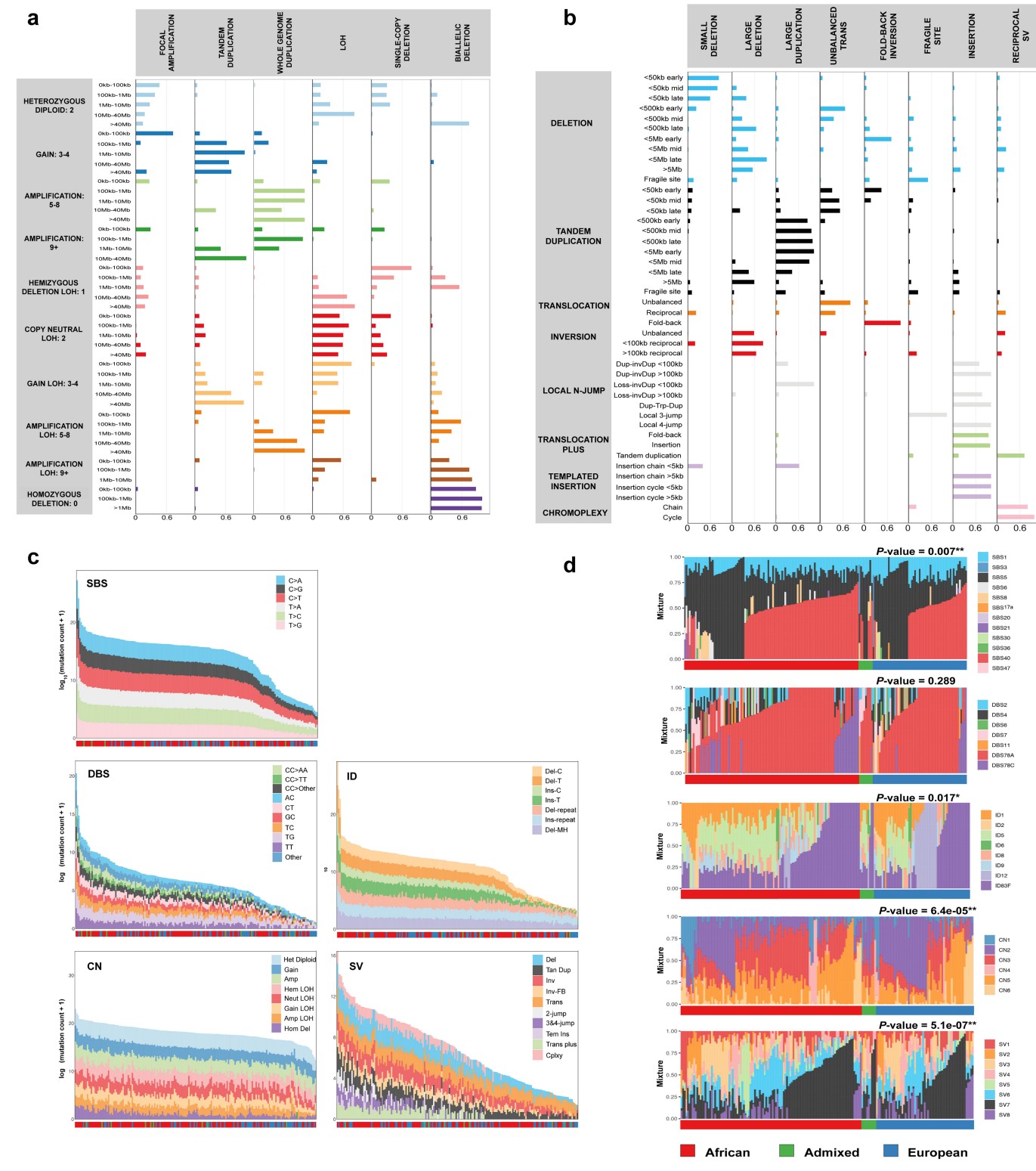

**Extended Data Fig. 8** | See next page for caption.

**Extended Data Fig. 8 | Known and novel mutational signatures in prostate cancer. a**, Copy number signatures in prostate cancer across 45 CN features ranked by mutational processes observed. The six most distinctive signatures and their important components extracted by the NMF algorithm were run on the sample size of 183 genomes. Bar charts represent the estimated proportion of each event feature assigned to each signature (rows sum to one). **b**, Structural variation signatures in prostate cancer ranked by mutational processes observed from small deletion to reciprocal rearrangement. The eight most distinctive signatures and their important components extracted from 44 features using the NMF algorithm were run on the sample size of 183 genomes. Bar charts represent the estimated proportion of each event feature assigned to each signature (rows sum to one). **c**, Frequency of SBS, DBS, ID, CN and SV features across 183 tumours. Colours at the bottom panel show the following ancestral groups: *i*) African, red; *ii*) Admixed, green; and *iii*) European, blue. **d**, Stacked barplots of multiple signature exposures for each mutational type enriched per patient and ranked by ancestral group. In many cases, certain mutational signatures occur more frequent in a tumour than others. The top enrichment of small- to large-size mutational signatures mentioned is shown for each patient in Supplementary Table 9 (see Enrichment). Copy number and structural variation signatures (CN1-6 and SV1-8, respectively) are the first identified in this study for prostate cancer, and their top enrichment of signature mixture/exposure per patient appears to be significantly associated with our GMS (one-way ANOVA or Fisher's exact test, two-sided *P*-values = 5.1e-07–0.017), considering either de novo or global mutational signatures discovered in the Catalogue of Somatic Mutations in Cancer (COSMIC). This supports a role of GMS in explaining intrinsic and extrinsic mutational processes in cancer.

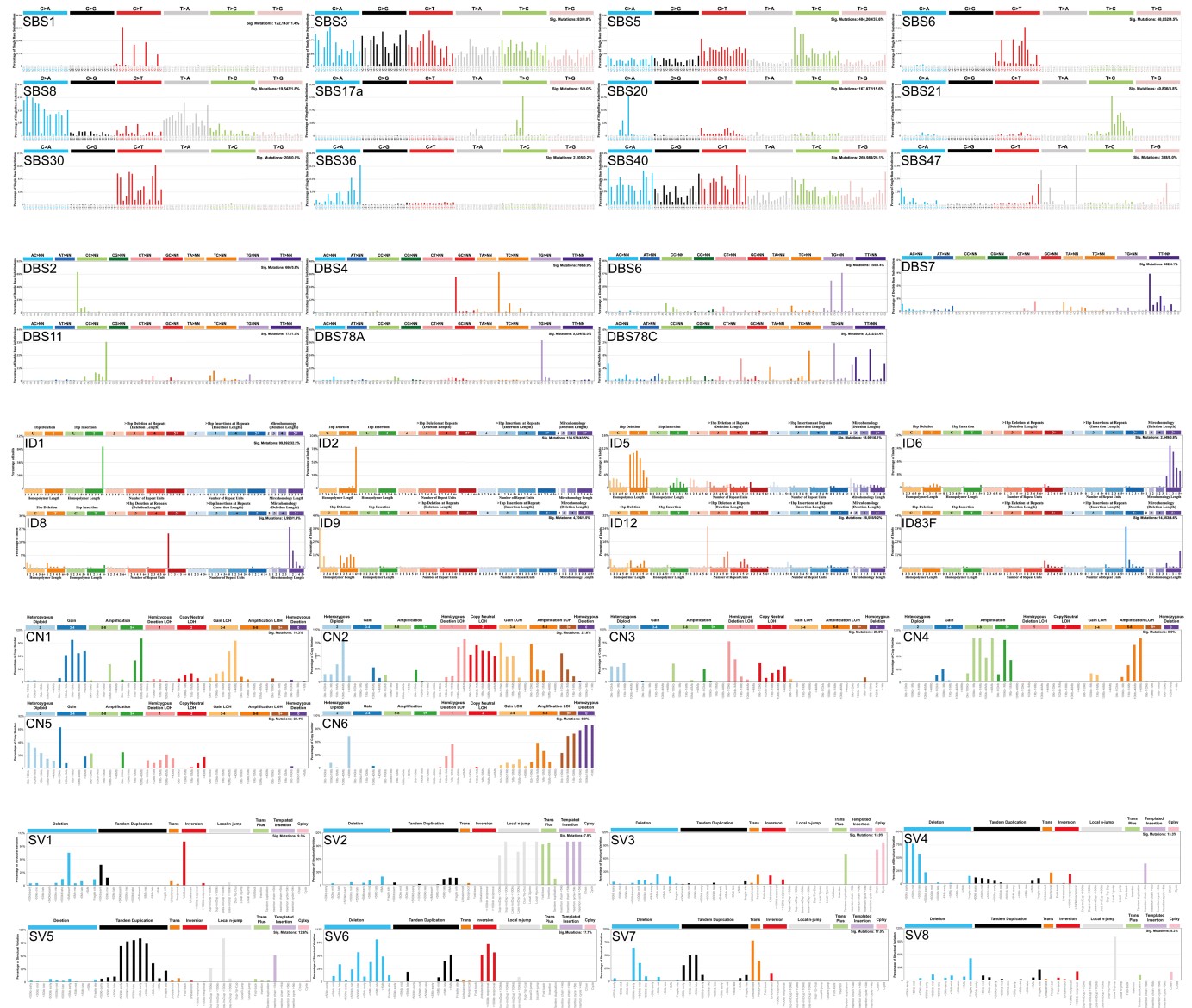

**Extended Data Fig. 9 | Total profiles of SBS, DBS, ID, CN and SV signatures.**
The classification of each signature type (SBS, 96 classes; DBS, 78 classes; ID, 83 classes; CN, 45 classes; and SV, 44 classes) is described in Supplementary Methods. The plotted data are available in digital form (Supplementary Table 9).

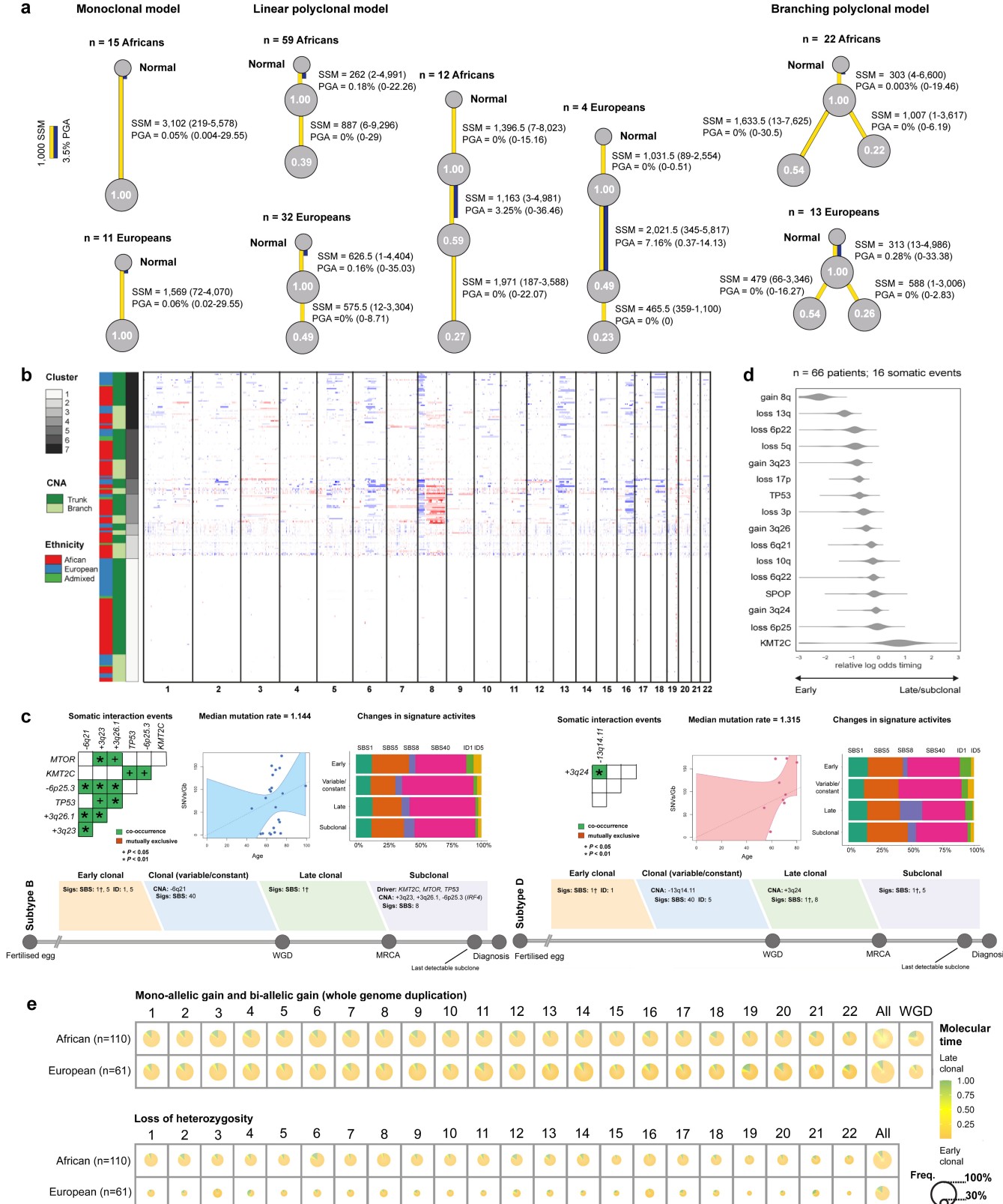

**Extended Data Fig. 10** | See next page for caption.

**Extended Data Fig. 10 | Stages of prostate tumour development. a**, Clonal architecture and its frequency in prostate cancer between Africans and Europeans. Tumours are divided into three groups: monoclonal, linear and branching polyclonal. The number of small somatic mutations (SSM) and CNAs as percentage of genome alteration (PGA) is provided as median and range in bracket. Cancer cell fraction (CCF) in each clone and/or subclone is shown in a circular node. Tumours that show characteristics consistent with being polytumours or with multiple independent primary tumors are excluded to remain conservative. **b**, Unbiased hierarchical clustering of CNAs between clonal (trunk) and subclonal (branch) mutations. Trunk mutations encompass those that occur between the root node (normal) and its only child node, while all others are classified to have occurred in branch. Red indicates gain; blue indicates loss; and rows indicate patients. Unidentified regions in trunk and branch are assumed to have neutral copy number. ConsensusClusterPlus showed seven CNA clusters among our patients to be optimal. The figure shows that a trunk alteration from one patient is mutationally similar to a branch alteration from another, rather than to other trunk ones from different patients in a cohort. **c**, Cancer timelines of GMS-B and -D identified in this study.

Detailed explanation is provided in Fig. 5. Significant somatic interactions based on Fisher's Exact test are indicated by odds ratio (OR) estimates and two-sided $P$-values on the top left panels. Interaction significance between somatic events in GMS-B and -D has $P$-values ranging from 3.16e-22–0.041 and 9.11e-25, respectively. Mutation rate plots show the median ±2× standard error of fitted data as dashed lines and error bands, respectively. **d**, Relative ordering model (PhylogicNDT LeagueModel) results for a cohort of 66 samples. The samples can be analysed if they have somatic events of interest prevalent greater than 5% of the sample size and have informative clonal status available for each event (16 events). Probability distributions show the uncertainty of timing for specific events in the cohort. **e**, Molecular timing distribution of copy number gains and loss of heterozygosity (LOH) between Africans and Europeans. Pie charts depict the distribution of the inferred mutation time for a given copy number alteration. Orange denotes early clonal gains/LOH, with a gradient to green for late gains/LOH. The size of each chart is proportional to the recurrence of this event across different patients. Most of the gains and LOH are considered early clonal based on MutationTimeR results. Whole-genome duplication is more frequent in Africans (63%) than in Europeans (57%).

# Reporting Summary

## Statistics

For all statistical analyses, confirm that the following items are present in the figure legend, table legend, main text, or Methods section.

| n/a | Confirmed | |
|---|---|---|
| ☐ | ☒ | The exact sample size (*n*) for each experimental group/condition, given as a discrete number and unit of measurement |
| ☐ | ☒ | A statement on whether measurements were taken from distinct samples or whether the same sample was measured repeatedly |
| ☐ | ☒ | The statistical test(s) used AND whether they are one- or two-sided<br>*Only common tests should be described solely by name; describe more complex techniques in the Methods section.* |
| ☐ | ☒ | A description of all covariates tested |
| ☐ | ☒ | A description of any assumptions or corrections, such as tests of normality and adjustment for multiple comparisons |
| ☐ | ☒ | A full description of the statistical parameters including central tendency (e.g. means) or other basic estimates (e.g. regression coefficient) AND variation (e.g. standard deviation) or associated estimates of uncertainty (e.g. confidence intervals) |
| ☐ | ☒ | For null hypothesis testing, the test statistic (e.g. *F*, *t*, *r*) with confidence intervals, effect sizes, degrees of freedom and *P* value noted<br>*Give P values as exact values whenever suitable.* |
| ☒ | ☐ | For Bayesian analysis, information on the choice of priors and Markov chain Monte Carlo settings |
| ☐ | ☒ | For hierarchical and complex designs, identification of the appropriate level for tests and full reporting of outcomes |
| ☐ | ☒ | Estimates of effect sizes (e.g. Cohen's *d*, Pearson's *r*), indicating how they were calculated |

*Our web collection on statistics for biologists contains articles on many of the points above.*

## Software and code

Policy information about availability of computer code

| Data collection | Data and metadata were collected from International Cancer Genome Consortium (ICGC), using ICGC Data Portal search (https://dcc.icgc.org/). Data repositories specific to CPGEA were also used for data collection (http://www.cpgea.com). |
|---|---|
| Data analysis | The core computational pipelines used in this study for read alignment, quality control and variant calling are available to the public at https://github.com/Sydney-Informatics-Hub/Bioinformatics. Analysis code for chromothripsis and chromoplexy is available through another GitHub page, https://github.com/tgong1/Code_HRPCa. Individual software components are as follows: fastp v.0.20.0; bwa v0.7.15; bwakit v. 0.7.17; SAMbamba v.0.7.1; SAMblaster v. 0.1.24; SAMtools v. 1.10; GATK v 4.1.2; ShatterSeek v0.4; ChainFinder v1.0.1; fastSTRUCTURE v1.0; Pophelper v2.2.7; ActiveDriverWGS v1.0.1; GISTIC v2.0.23; CNVkit v0.9.6; Manta v1.6.0; GRIDSS v2.8.3; fishHook v0.1; Bionano Access 1.5.2; ConsensusClusterPlus v1.50.0; ActivePathways v1.0.2; TitanCNA snakemake workflow v1.17.1. |

For manuscripts utilizing custom algorithms or software that are central to the research but not yet described in published literature, software must be made available to editors and reviewers. We strongly encourage code deposition in a community repository (e.g. GitHub). See the Nature Portfolio guidelines for submitting code & software for further information.

## Data

Policy information about availability of data

All manuscripts must include a data availability statement. This statement should provide the following information, where applicable:

- Accession codes, unique identifiers, or web links for publicly available datasets
- A description of any restrictions on data availability
- For clinical datasets or third party data, please ensure that the statement adheres to our policy

DNA sequence data have been deposited in the European Genome-Phenome Archive (EGA; https://ega-archive.org) under overarching accession EGAS00001006425 and including the Southern African Prostate Cancer Study (SAPCS) Dataset EGAD00001009067 and Garvan/St Vincent's Prostate Cancer Database EGAD00001009066. Academic researchers meeting the Data Access Policy criteria, may apply for data access via the respective Data Access Committees. CPGEA data are available via http://www.cpgea.com. PCAWG data are available at ICGC Data Portal (https://dcc.icgc.org/releases/PCAWG).

March 2021

# Field-specific reporting

Please select the one below that is the best fit for your research. If you are not sure, read the appropriate sections before making your selection.

☒ Life sciences   ☐ Behavioural & social sciences   ☐ Ecological, evolutionary & environmental sciences

For a reference copy of the document with all sections, see nature.com/documents/nr-reporting-summary-flat.pdf

# Life sciences study design

All studies must disclose on these points even when the disclosure is negative.

| | |
|---|---|
| Sample size | Sample sizes were determined in order to obtain nearly 200 tumour-blood pairs, based on biospecimen availability, with a focus on samples of underrepresented populations. All primary tumours and matched-blood tissue from 190 specimens were used to generate sequencing data in this study. We considered this sample size would be sufficient because our significant comparison of tumour genome profiling between Europeans (n=9) and Africans (n=6) has been previously published in a peer-reviewed journal. For comparisons, 93 CPGEA donors were included due to high-risk prostate cancer with most treatment-naive and 628 PCAWG donors were chosen based on different primary ancestries. Additional 256 prostate cancer patients from PCAWG were compared with most treatment-naive. |
| Data exclusions | After quality assurance, data from 8 tumour-blood pairs were excluded as unusable. Reasons for data exclusion included evidence of cross-contamination and duplication. Hypermutated tumours (30 mutations/Mb) were removed in mutational recurrence analysis of small mutations and cancer evolution analysis , following ActiveDriverWGS and PhyloWGS software user manuals. |
| Replication | The accuracy of SV breakpoint inference was assessed by applying two different algorithms and selecting only calls detected by both. Integrative clustering analysis was re-assessed using independent clustering of each dataset, with subsequent results mostly recapitulating the subtypes found by the integrative analysis. |
| Randomization | N/A - This exploratory study of genome profiling tumours in underrepresented populations did not contain a randomisation step due to biospecimen scarcity. |
| Blinding | N/A - This exploratory study within underrepresented populations did not contain a blinded data collection due to the focus on those populations in this study. Sequencing and early steps in data analysis were partially blinded using a pool of all samples of different ancestries collected. |

# Reporting for specific materials, systems and methods

We require information from authors about some types of materials, experimental systems and methods used in many studies. Here, indicate whether each material, system or method listed is relevant to your study. If you are not sure if a list item applies to your research, read the appropriate section before selecting a response.

## Materials & experimental systems

| n/a | Involved in the study |
|---|---|
| ☒ | ☐ Antibodies |
| ☒ | ☐ Eukaryotic cell lines |
| ☒ | ☐ Palaeontology and archaeology |
| ☒ | ☐ Animals and other organisms |
| ☐ | ☒ Human research participants |
| ☒ | ☐ Clinical data |
| ☒ | ☐ Dual use research of concern |

## Methods

| n/a | Involved in the study |
|---|---|
| ☒ | ☐ ChIP-seq |
| ☒ | ☐ Flow cytometry |
| ☒ | ☐ MRI-based neuroimaging |

# Human research participants

Policy information about studies involving human research participants

| | |
|---|---|
| Population characteristics | Patient-by-patient clinical data are provided in Supplementary Table 1.  Demographically, the cohort included 53 Australians, 7 Brazilians and 123 Africans, with ages ranging from 45-99 years old (median 65.5 yo). Having performed genetic ancestry analysis, the cohort consisted of 113 Africans, 61 Europeans and 9 Admixed mostly between African and European. Preoperative PSA levels ranged from 3.5 to 4,847 ng/ml (median 22.9 ng/ml). ISUP Grade Groups were distributed as follows: 0-2: 29 (16.6%); 3: 11 (6.3%); 4: 52 (29.7%); and 5: 83 (47.4%). All patients are male. |
| Recruitment | After obtained the consent of patients, 183 patients from Australia (n=53), Brazil (n=7) and South Africa (n=123) and presenting mostly with clinicopathologically confirmed prostate cancer had their tumour and blood samples collected. All except one Australian patient (PID 15178) treated with one-month-long Ozurdex therapy were treatment naïve at time of sampling. Three patients were unconfirmed for the cancer and confirmed for benign prostate hyperplasia (BPH). All men from the Southern African Prostate Cancer Study (SAPCS) were recruited at the time of diagnosis, and therefore tumour tissue was derived from biopsy core, while age and PSA levels were recorded at the time of diagnosis. Australian and Brazilian subjects were recruited at the time of radical prostatectomy. Additional selection criteria included: availability of fresh-frozen tissue and matched blood, self-reported ethnicity and country of origin, as well as availability of clinical and pathological data. <br><br> Results in this study would not represent all underrepresented populations in Africa and South America as only South Africa and Brazil were studied. |

Ethics oversight

All samples were obtained with written informed consent, as per study approval granted from the St. Vincent's Human Research Ethics Committee in Australia (HREC), SVH/12/231, the Grupo de Pesquisa e Pós-Graduação (GPPG) Scientific Committee and Research Ethical Commission (IRB) approval number 20160539 in Brazil or the University of Pretoria Faculty of Health Sciences Research Ethics Committee (with US Federal wide assurance FWA00002567 and IRB00002235 IORG0001762) approval number 43/2010 in South Africa. Samples were shipped to the Garvan Institute of Medical Research in accordance with institutional Material Transfer Agreements (MTAs), as well as additional Republic of South Africa Department of Health Export Permit (National Health Act 2003, J1/2/4/2 No 1/12). Whole genome sequencing and analysis were performed in accordance with approval granted by St. Vincent's Hospital HREC SVH/15/227 and governance review authorisation granted for human research at the Garvan Institute of Medical Research GHRP1522.

Note that full information on the approval of the study protocol must also be provided in the manuscript.

