## [Peer Review File · Nature]

Manuscript Title: African-specific prostate cancer molecular taxonomy

Reviewer Comments & Author Rebuttals

Reviewer Reports on the Initial Version:

Referees' comments:

Referee #1 (Remarks to the Author):

This is an important first extensive study of whole genomes of treatment-naive prostate cancers from South African men, notably including 113 African ancestry men, a group that has largely been uncharacterized at the genomic level. Men of African ancestry suffer a disproportionate burden from prostate cancer. The major results of this study include the generation of whole genome sequencing data of prostate cancers from South African men including 113 men of African ancestry. The authors perform an extensive analysis and use these data alongside existing whole genome sequencing data from men of other ancestries to report a new classification system of prostate cancers that identifies African-specific prostate cancers.

There are a number of suggestions I would recommend.

Major points include revision of figures and additional analyses to highlight the major findings of the study.

1) Currently the presentation of the data in figures 1 and 2 render the findings more challenging to interpret and draw less attention to the major findings of the study. Could the authors highlight the major driver genes and alterations in this cohort by European and African Ancestry, and providing a clearer comparison between the findings of this study in terms of mutated genes, copy number alterations, in comparison between these data and TCGA/PCAWG (European, African American) which they reference. In addition, please indicate the frequency of the mutations in the text and comment on the prevalence of these alterations. While there is a reason to group all 183 together in one cohort, one major question in the field is whether and how somatic mutations differ by ancestral group and this area could be more substantially focused on as one major advancement is the inclusion of whole genome African ancestry prostate cancers.

2) Figure 3 displays the subtyping of the cohort. Panel A orders the genes by significance value – I think this type of analysis is somewhat difficult to follow and interpret given the different color coded categories provided without accompanying prevalence information. Could the authors clearly describe how these subtypes are generated and how they are distinguished from previous subtyping? The columns show gene names but there are multiple columns for each gene name (Panel B). Please clarify and also add missing labels to subtypes. A clearer schematic of the subtyping schema/features could be helpful here. Subtype B appears to show many copy number gains across the specified genes; please comment and provide further analysis given that this is proposed to be an African-specific subtype.

3) Overall, more analysis and explication would be very helpful to support and define the identity of the "African-specific genomic subtype" as described by the authors. How do these new global subtypes compare to known molecular subtypes, what are they comprised by in terms of known molecular subtypes/mutational events?

4) Within the European vs African comparisons, what CNA and genomic alterations are significantly different that could inform our understanding of the biology?

5) Another major question and potentially interesting analysis is how African prostate cancers compare with African American prostate cancers – could the authors provide some comparison to these data? Could the authors comment on other genes/alterations previously identified as

potentially associated with ancestry (LSAMP deletion, SPOP, ERF etc) Could the authors analyze the germline for any DNA repair mutations or other known risk alleles? It would likely be informative to know.

6) For the comparisons that are found to be statistically significant between the European and African samples such as PGA, mutations per samples, other relevant comparisons, how do the Gleason grades inform their interpretation? If there are higher Gleason grade tumors among the African cases, how do the authors think this affects their comparisons? There appear to be some hyper mutated tumors in the African ancestry cohort which could affect the comparisons.

Minor points

Overall several analyses were described though p-values were apparently non-significant.

1) In lines 115-117 seemed unclear why nonsignificant findings were described as such Chromoplexy was more frequent in Europeans than in Africans (38% versus 33%, P-value=0.536), with the number of interchromosomal chains more likely to be elevated in Africans than Europeans (1-6 versus 1-2, P-value=0.748).

2) Listing by FDR is likely somewhat less informative by itself and could be bolstered by indicating the prevalence of significant CNAs. Please comment on why this and consider directly showing and comparing prevalences.

4) Authors indicate in lines 124-125 about "probably and possibly damaging, were significantly greater in Africans (PolyPhen-2, 14 versus 11 mutations in Europeans, P-value=0.022, two sample t-test)" - is this per patient? please specify .

5) In lines 159-161 it is again noted a difference in prevalence of SPOP coding mutations but the p-value is 0.426.

6) In Figure 3 there are complete blue bars that extend throughout several gene columns – what does this signify?

7) In Ext Figure 4A, the figure legend doesn't list the overall cohort numbers or which exact cohort of patients this panel is characterizing.

8) In Fig Ext Data 4, it would be helpful to please specify the cohorts being referred to, such as the total number within the group.

9) In Ext Data Figure 5, the authors show a number of affected pathways/genes across the entire group of 183, though it appears some alterations are restricted or enriched in one group; comparisons within the group of significant genes or to other cohorts would be informative.

10) In Fig Ext Data 8 Panel D, the authors state "Copy number and structural variation signatures (CN1-6 and SV1-8, respectively) are the first identified in this study for prostate cancer, and their enrichment in a patient appears to be significantly associated (P-values <0.05) with our GMS." Please clarify how these exact signatures appear and are defined and are associated; how do the authors interpret this?

11) Ext Data Figure 2 which depicts the top 300 genes altered across the cohort was somewhat difficult to interpret given only some gene names listed (TP53 is top but then no label for several rows below it that also show high number of losses); also missing a label of right panel (total counts?). As these could be considered two sub cohorts within the cohort, it could be helpful to also depict comparisons directly between the two cohorts as bar plots.

Referee #2 (Remarks to the Author):

This manuscript appears to present findings from the largest and only study of prostate cancer genomes from patients in Sub-Saharan Africa. This is a highly significant and novel study since prostate cancer genomics studies are grossly underrepresented among patients of African ancestry. There are no concerns regarding the approach, quality of data and its presentation. The appropriate statistical tests appear to be used throughout the analysis and conclusions are appropriate.

Referee #3 (Remarks to the Author):

In the manuscript by Jaratlerdsiri et al, novel whole-genome sequencing data of prostate tumors is presented that uses somatic variant calling to derive mutational subtypes on a global scale. The major conclusion presented by the evidence in this paper is that the comparison of the subtypes can explain the development of tumors over a timeline comprising the natural life of the patient, due in part to differing mutation rates.

The most novel and compelling component to this paper are the 123 tumor whole-genomes from South African patients. This is itself an amazing resource that is of tremendous value to population scientists and cancer researchers. Strengths of this work include a robust comparison to other datasets (including PCAWG), and the bioinformatic analyses are both well-described and thorough.

Overall, however, I have general and specific concerns that decrease my enthusiasm about the broad conclusions argued.

1) The vast majority of descriptive analyses in the first 2/3 of this work compare the whole-genome analyses of South African patients versus Australian and Brazilian patients. A quick review of Supplementary Table 1 shows significant differences in these groups, including the use of a biopsy sample (SA) vs prostatectomy (BR and AU) and their baseline PSA levels (239 vs 15). This alone argues that the SA cohort is potentially higher risk, and if any subset of these patients had metastatic disease (which is highly likely) the age/timing of the disease is different as is the possibility of metastatic re-seeding that could influence downstream analyses. If any of these patients had metastatic disease the conclusions of this work would be different.

2) Despite these clinical differences, the objective mutational profiles between their cohorts (as presented in Figures 1 and 2) do not show much distinction between geographic groups. Even when limiting to drivers from unbiased analyses, integrative clustering does not demonstrate compelling differences. While this still is in alignment with the manuscript's goal of presenting a molecular taxonomy, it does not change the current views of prostate cancer from previous datasets.

3) Some discussion of alterations and driver genes are confusing. Data presented in the text and in Figure 3 suggest that some alterations occur in 100% of samples (line 173). Please consider editing the methods to validate or curate these striking findings.

4) The initial presentation of the GMS classifier is itself confusing. Is it supposed to be presented in Figure 3b? (The manuscript says 2b.) Are each group of rows/boxes the GMS A-D? Consequently, interpretation of data based on the GMS scores (including the SBS signatures) do not reference individual GMS groups, so I struggled to appreciate the significance of the data presented in Figure 4 and how it related to these GMS groups.

5) The molecular evolution and timing data presented in Figure 5 is quite strong and strengthened by the Extended data. However, the confidence intervals around mutation rates (5a and 5b) are extremely large with respect to patient age. To what extent could these differences be driven by the higher risk clinical features of the South African group?

Minor concerns:

1) The paper refers to their cohort as Sub-Saharan, African and South African. Supp Table 1 indicates all patients were from South Africa, the country. As Sub-Saharan Africa comprises over 40 countries, this language should be clarified.

2) Some of the intervals presented are non-sensical. For example, Line 127 says a median of 2+/-22.5 coding drivers was observed. Since -20.5 coding drivers cannot occur, perhaps a different

two-sided analysis should be used accepting a distribution that is left-censored at zero?

Referee #4 (Remarks to the Author):

Hayes and colleagues conduct an important study in the WGS of sub-Saharan African genomes of men with prostate cancer and compare these to non-African prostate cancers. This is the first study which addresses African continent prostate cancers which may of course be different from African American or Afro-Caribbean prostate cancers-studies the authors argue for in later studies. They point out important differences in DNA mutations, cancer drivers and come up with a novel taxonomy that groups their cohort into 4 groups-one that is highly ethnically mixed and 2 that appear to be African specific.

They further use clonal evolution analyses to associate clock-like processes to mutation acquisition and infer differential causality based on signature analyses in the African genomes.

The work is novel. The implications for prostate cancer etiology and personalised medicine approaches could be very important.

My major initial concern is the initial cohort itself in the comparison between the caucasian and African clinic-pathologic characteristics. Extended Figure 1a should be modified and statistically compared between African and non-African patients such that there is not bias in the characteristics between the two groups. If the clinical factors are more aggressive in the African group, then relative already- known entities such as PGA, mutations, drivers etc can be explained by higher GG or PSA or T-category or intraductal carcinoma. This is less of an issue when there are novel findings for African genomes, but very important for the relative calling of mutations of African vs non-African.

In some instances, the authors describe "significant" increases in mutations in African genomes with supporting p-values, but it can be perceived as slightly disingenuous to say "increased" with a non-significant p-value. Even if an absolute value is larger, the authors should rather state "no difference" if the statistics do not support a difference. Otherwise, there is a subconscious bias in the reader that more differences exist than actual testing supports.

A small point is the use of the PGA values - this is somewhat confusing - can the authors use percentages for PGA (e.g. 0.078; is this not 7.8% rather than 0.078%? The reader could be confused relative to PGAs from 1-30% in localised disease reported in other papers..

Can the authors be more explicit in terms of differences between African versus African-American genomes ? There must be some WGS carried out-certainly there are mutation spectra in AACR GENIE that have been published in NEJM between African-American, Caucasian and Asian populations-this might strengthen their case.

The argument around SBS, ID and other signatures in African versus non-African and across GMS groups A-D is unclear in the manuscript-what is the interpretation regarding ethology of the sub-groups when this is applied.

Can the authors be explicit surrounding the timing of the chromosomal gains and losses within African versus non-African genomes ?

The pathway analyses are important and do relate to the TCGA pathways published, but the gene-specific reasons for the pathways are non clear. For example, when showing data in which DDR - genetic instability is increased in African versus non-African, what genes or processes are driving this ? Is it mismatch repair defects as evidenced from the signature analyses (as the HR signatures

do not seem to be as important) or specific genes - I note that this study would suggest that BRCA2 is not a driver in Africa? Is that the case ?

For clarity and context to a generalist audience for Nature or Nature journals, the abstract should be placed into more of a lay context. A summary table at the end of the manuscript pointing out the major differences between African and non-African prostate genomes would be very helpful and I suspect off quoted and shown in many slide sets. I

Response to Referee's Comments

African-specific prostate cancer molecular taxonomy

Nature manuscript 2021-11-17434A. Jaratlerdsiri W, et al., Hayes V.M.

Note: All changes to the *Main Text*, in response to referee's comments are highlighted in **red** for ease of recognition.

Referee #1 (Remarks to the Author):

Comment: This is an important first extensive study of whole genomes of treatment-naive prostate cancers from South African men, notably including 113 African ancestry men, a group that has largely been uncharacterized at the genomic level. Men of African ancestry suffer a disproportionate burden from prostate cancer. The major results of this study include the generation of whole genome sequencing data of prostate cancers from South African men including 113 men of African ancestry. The authors perform an extensive analysis and use these data alongside existing whole genome sequencing data from men of other ancestries to report a new classification system of prostate cancers that identifies African-specific prostate cancers.

Response: We appreciate Referee #1 recognising the lack of African-relevant genomic data, specifically from Africa, while highlighting the significant disproportionate prostate cancer burden for men of African ancestry. He/she also takes note of our extensive analysis performed and comparative analyses between ethnicities, which led to the identification of African-specific subtypes of particular relevance to aggressive disease.

Comment with different Points: There are a number of suggestions I would recommend. Major points include revision of figures and additional analyses to highlight the major findings of the study.

Point 1) Currently the presentation of the data in figures 1 and 2 render the findings more challenging to interpret and draw less attention to the major findings of the study. Could the authors highlight the major driver genes and alterations in this cohort by European and African Ancestry, and providing a clearer comparison between the findings of this study in terms of mutated genes, copy number alterations, in comparison between these data and TCGA/PCAWG (European, African American) which they reference. In addition, please indicate the frequency of the mutations in the text and comment on the prevalence of these alterations. While there is a reason to group all 183 together in one cohort, one major question in the field is whether and how somatic mutations differ by ancestral group and this area could be more substantially focused on as one major advancement is the inclusion of whole genome African ancestry prostate cancers.

Response: Our study was critically designed to achieve a comparative analysis between an African and European ancestral treatment-naive largely aggressive disease presenting prostate cancer cohort where tumour-blood paired whole genome data is generated and interrogated using a single technical and informatic pipeline. While **Figure 1** addresses the overall mutational density (by somatic variant type) between the ancestries, we agree with Referee 1 that we should pay closer attention to ancestral differences (a significant question in the field and focus of this paper), and as such, we have added two additional components to **Figure 2**, highlighting the differences in driver alterations by ancestral group.

New Figure 2, with added/altered text in red.

Fig. 2 | Prostate cancer driver mutation differences and novel taxonomy by ancestry. **a**, The selected 35 driver genes classified as either; *i*) most altered in this study (>10 patients), irrespective of ancestry (green); *ii*) DNA Damage Repair (DDR) genes known to associate with African ancestry (orange); *iii*) other ancestry-associated genes studied in prostate cancer (purple); or the odds ratio (OR), 95% confidence interval, and *P*-value (<0.05) were calculated using Fisher's Exact test for count data and including 10 African-specific (OR = 0) and 3 European-specific (OR = infinity) genes. **b**, Dual barplot illustrating the mutational frequency of the altered driver genes between Africans and Europeans by mutational type (CDS, coding sequence; SV, structural variation; and CN, copy number).

Additional relevant text has also been added at original submission, which directly responds to Referee 1's comments and refers to the new Figure 2a and 2b. The added text can be found from **Line 154** in the **Main Text 'Rev1'** and reads as follows, with the inclusion of an additional reference [24]. **Added text:** Differences in oncogenic driver alterations between ethnicities were observed (Fig. 2a, b). Specifically, African-derived tumours were more likely to have CNAs and mutations in *SETBP1* (Frequency=0.33, OR=0.357, *P*-value=0.012), *DDX11L1* (0.48, OR=0.24, *P*-value=0.0001), *STK19* (0.25, OR=0.215, *P*-value=0.004), and *NCOA2* (0.51, OR=0.172, *P*-value=3.14e-06), along with SVs in *PCAT1* (0.13, OR=0.11, *P*-value=0.012). Conversely, SVs for *TMPRSS2* (0.38, OR=3.639, *P*-value=0.0006) and *ERG* (0.34, OR=3.159, *P*-value=0.003) were more notable among Europeans. Although several DNA Damage Repair (DDR) genes and others previously associate with African ancestry were not significantly altered between Africans and Europeans in this study, 10 were solely altered in Africans with most in the coding sequence (CDS; Frequency, 0.009-0.035). All these support the inclusion of a larger number of underrepresented populations in clinical enrolment for the benefit of precision oncology studies²⁴.

Added reference:

24 Mahal, B. A. *et al.* Racial Differences in Genomic Profiling of Prostate Cancer. *N Engl J Med* **383**, 1083-1085 (2020).

In response to directly comparing our data with other cohorts for racial disparity, with Referee 1 specifically mentioning TCGA and PCAWG and data of relevance to African Americans, direct comparison was not possible due to the following reasons. For **PCAWG**, of the 199 PCa patients, only eight were classified as African American, while **TCGA** reports data for 427 patients of which nine have been classified as African (>90% ancestral threshold) and 61 as Admixed (assumed African American with <90% African ancestral threshold). The major limitation to merging TCGA data with our study is that in contrast to our WGS data (from which our taxonomy was derived), TCGA only provides somatic exomic mutations and general copy number data (not unbiased set of copy number drivers) and no structural variation. In response to **Referee #4** we compare our data to **AACR GENIE** (discussed later).

Point 2) Figure 3 displays the subtyping of the cohort. Panel A orders the genes by significance value – I think this type of analysis is somewhat difficult to follow and interpret given the different color coded categories provided without accompanying prevalence information. Could the authors clearly describe how these subtypes are generated and how they are distinguished from previous subtyping? The columns show gene names but there are multiple columns for each gene name (Panel B). Please clarify and also add missing labels to subtypes. A clearer schematic of the sub typing schema/features could be helpful here. Subtype B appears to show many copy number gains across the specified genes; please comment and provide further analysis given that this is proposed to be an African-specific subtype.

Response: The clarify the questions addressed by Referee 1, we updated figure 3 (see below). Most specifically, through merging old Fig 3a and b (new Fig. 3a), we provide prevalence information and clarification on how the subtypes were generated and distinguished from previous subtypes, while providing additional information through expanding the schematic labelling. While the original text comments on the excess CN gains in Subtype B, see sentence from text which states, “Subtype D showed the greatest mutational density (1.91 mutations/Mb, 1.08 breakpoints/10Mb, 31% PGA) with a mixture of copy number (CN) gains and losses, while Subtypes B and C were marked by substantial CN gains or losses, respectively (Fig. 3a)”, additional reference to Subtype mutational pattern differences are further highlighted in figure 2 legend for further clarification.

New Figure 3 and new Supplementary Table 7, with added/altered text in red.

Fig. 3 | Significance in somatic aberrations across four diverse subtypes. **a**, Analysis of the long tail of driver genes using different combinations of mutational types (CDS, coding driver data; NC, noncoding driver data; SV, significantly recurrent breakpoint data; and CN, gene-level copy number data) resulting in the identification of 124 preferentially mutated genes among the subtypes. Ordered by mutational frequency, 100 (80.6%) have been reported as significantly recurrent mutations/SV breakpoints in the PCAWG Consortium²², while 24 (19.4%) are significantly mutated in this study (marked by asterisks). Using iClusterplus, unsupervised hierarchical clustering of all mutational types identified four prostate cancer subtypes (A-D, Fig. 2c), presented for 183 patients (rows) and 124 mutated genes (columns), with each subgroup ordered by ethnicity. Ethnically diverse Subtypes A and C are mutationally quite or marked by CN loss, respectively. African-specific/predominant Subtypes B and D are marked by CN gains and mutationally noisy, respectively. Three genes on chromosome X, *KDM6A*, *ATRX* and *ZMYM3* are considered significant due to the abundance of homozygous loss present in Subtype C. **b**, Kaplan-Meier plot of biochemical relapse (BCR)-free survival proportion of European patients in subtype A (n=161) versus C (n=19). **c**, Kaplan-Meier plot of cancer survival probability of European patients in subtype A (n=82) versus C (n=17).

Point 3) Overall, more analysis and explication would be very helpful to support and define the identity of the “African-specific genomic subtype” as described by the authors. How do these new global subtypes compare to known molecular subtypes, what are they comprised by in terms of known molecular subtypes/mutational events?

Response: To provide further analysis and explanation that defines the different Subtypes, especially relevant for comparing the African-specific/predominant (GMS-B and -D) versus the Universal or ethnically diverse Subtypes (GMS-A and -C), in Figure 4 (adding Figure 4b), we provide additional analysis that thoroughly compares the Mutational Signatures between the Subtypes and further highlight the unique features of the African-specific GMS-B. Associated text has been added in the *Main Text Rev1* from *Line 276* and reads, *Fig. 4b shows the duplication signatures to have at least 1.5 times larger proportion of genomic aberrations in GMS-B, -C and -D than the universal GMS-A. Furthermore, the African-specific subtype GMS-B consisted of several CN4 and SV5 genomic aberrations composed predominantly of CN amplification (>5 copies and mainly >100 kb in length) and tandem duplication (<5 Mb in size occurred during early to late timing of DNA replication), respectively.*

New Figure 4, including additional analysis presented as 4b, with added/changed text in red.

Fig. 4 | Estimates of genomic aberrations contributed by each mutational signature. a, Correlation plots of total mutational signatures along with clinical and genomic characteristics. The size of each dot represents FDR values of Spearman correlation *P*-values using BH correction. The colours of each dot represent correlation coefficient (ρ). GMS is assigned as 1-4 for Subtypes A-D, respectively; African, Admixed and European are recorded as 1-3, respectively. The correlation of 32 significantly mutated genes in prostate cancer is shown in the X-axis. **b, Sankey diagram depicting a proportion of duplication signatures observed across cancer subtypes.** Duplication features, including amplification (Amp), translocation plus, local n-jump, templated insertion, amplification loss of heterozygosity (LOH), gain, tandem duplication, and gain LOH (see Extended Data Fig. 8a, b) are summed per subtype and equally weighted to 20. Links connecting between nodes (GMS, signatures, and features, respectively) have widths proportional to the total number of CN or SV features across all patients within each GMS to

which they belong. Note that we believe GMS-B is the identity of the “African-specific genomic subtype”.

Point 4) Within the European vs African comparisons, what CNA and genomic alterations are significantly different that could inform our understanding of the biology?

Response: Refer to *Point 1 Response* and *Main Text Rev1 Line 154*.

Point 5) Another major question and potentially interesting analysis is how African prostate cancers compare with African American prostate cancers – could the authors provide some comparison to these data? Could the authors comment on other genes/alterations previously identified as potentially associated with ancestry (LSAMP deletion, SPOP, ERF etc) Could the authors analyze the germline for any DNA repair mutations or other known risk alleles? It would likely be informative to know.

Response: Refer to *Referee #4* where we compare our data to *AACR GENIE* data, including African Americans (discussed later). In response to the germline data, due to the bulk of the data from this paper and the focus on the identification of a new taxonomy, through extensive somatic variant analysis, we did not believe that this paper would do justice to the also very important germline data. For the *DNA repair mutations*, a manuscript addressing this question is currently under review at *Genetics in Medicine*, which is also now being expanded to include global comparative analysis. For the *known risk alleles*, it made more sense (due to lack of study power related to risk allele studies), to merge this data with a GWAS (exomic) analysis which we have performed on a larger cohort of Southern Africans. This paper is under draft and due to be submitted shortly. Those papers reference to this article.

Point 6) For the comparisons that are found to be statistically significant between the European and African samples such as PGA, mutations per samples, other relevant comparisons, how do the Gleason grades inform their interpretation? If there are higher Gleason grade tumors among the African cases, how do the authors think this affects their comparisons? There appear to be some hyper mutated tumors in the African ancestry cohort which could affect the comparisons.

Response: The patient selection was based on the following criteria: (i) self-reported ethnicity (as African or European) and geography (South African or Australian, with the inclusion of 7 Brazilians), (ii) treatment-naïve at time of diagnosis/surgery (to avoid treatment associated hyper-mutation), and (iii) Gleason score >7 at diagnosis/surgery (to allow for the best possible comparative analysis based on clinicopathology of the patient at the time of tumour sampling. To avoid technical and computational differences related to sequencing technology and informatic workflows, all samples underwent from DNA extract to WGS to analysis under a single workflow and at a single timepoint, with sample mixing to avoid batch effects.

Addressing the Gleason grades of the study cohort, we have added text that highlights that the South African cohort overall presented with the lowest percentage of high-grade tumours (ISUP GG 4/5) in our study at 72.2% (83/115), which equates to 72.4% (76/105) of African ancestral South Africans with confirmed pathology, compared with 86.8% (46/53) for the European Australians.

As 100% of the South Africans and therefore 100% of the Africans were treatment-naïve at time of tissue retrieval, it is unlikely that the clinical management played a role in the hyper-mutated tumours. As all primary tumours are matched for advanced disease (through study bias towards ISUP GG 4 and 5) between the African and European men, we would strongly argue that comparison is feasible and the best possible apples-to-apples match. The only unavoidable caveat is the time of sampling, biopsy at diagnosis (all South Africans and all Africans) *versus* surgery (all Australians and most Europeans). As all Australians were treatment naïve at time of surgery, they were selected for patients who as close to time of diagnosis opted for surgery.

For further clarity regarding the points addressed above, additional information in red has been added to the *Main Text Rev1* and reads from *Line 72* as: “Controlling for study artefacts, an additional **53 Australians and seven Brazilians** were passed simultaneously through the same high-depth whole-genome sequencing (WGS), mutation-calling and analytical framework. Focusing on treatment-naïve cases (**100% South Africans, 98% Australians and two confirmed Brazilians**) and aggressive tumours (mostly Grades 4-5, **72.2% South Africans, 86.8% Australians, and 85.7% Brazilians**; Extended Data Fig. 1a) and patient-matched blood achieving coverages of 88.69 ± 14.78 and 44.34 ± 8.11 , respectively (median \pm s.d., Supplementary Table 1), we uniformly generated, called and assessed about 2 million somatic variants.” Under *Genetic ancestry* from *Line 97*, we add the following detail in red: “Ancestries were assigned using 7,472,833 markers as: African (n=113, **all South Africans**), with greater than 98% contribution; European (n=61, **53 Australians, 5 South Africans, and 3 Brazilians**), allowing for up to 10% Asian contribution (with a single outlier of 26%); and African-European Admixed (n=9, **5 South Africans, and 4 Brazilians**), with as little as 4% African or European contribution (Extended Data Fig. 1b).”

Further with regards to the *hyper-mutated African-derived tumours*, we found that after exclusion of tumours with a TMB >30 mutations/Mb, we still observed a significant elevated TMB in our African over European-derived tumours, adding the following red text in *Main Text Rev1* from *Line 109*: “African derived tumours harboured a higher rate of small mutations (SNVs and indels), with a median of 1.197 mutations/Mb (**range 0.031-170.445**), compared to those of Europeans (1.061 mutations/Mb, P -value=0.013, two-sample t-test; **exclusion of hypermutated tumours at >30 mutations/Mb, P -value=0.028**). The same held true for the protein coding mutations (candidate oncogenic drivers), with the addition of the following text in red at *Line 126*: “Protein-coding mutations, including probably and possibly damaging, were significantly greater in **each** African (PolyPhen-2, 14 *versus* 11 mutations in a European, P -value=0.022, two-sample t-test; **exclusion of hypermutated tumours, P -value = 0.039**).

Comments on Minor points: Overall several analyses were described though p-values were apparently non-significant.

Point 1) In lines 115-117 seemed unclear why nonsignificant findings were described as such Chromoplexy was more frequent in Europeans than in Africans (38% versus 33%, P -value=0.536), with the number of interchromosomal chains more likely to be elevated in Africans than Europeans (1-6 versus 1-2, P -value=0.748).

Response: These non-significant findings have been removed and replaced with a shorter and more simplistic sentences that reads in the *Main Text Rev1* at *Line 117* as, “**No overall differences between the ethnicities were observed for chromothripsis (range 52-55%) and chromoplexy (range 33-38%), while African tumours demonstrated a trend towards a higher number of interchromosomal chromoplexic chains (1-6 versus 1-2).**”

Point 2) Listing by FDR is likely somewhat less informative by itself and could be bolstered by indicating the prevalence of significant CNAs. Please comment on why this and consider directly showing and comparing prevalences.

Response: Addressed in response to *Reviewer #1, Major Point 1 Response*, and presented in the changes made to *new Figure 2* and presented in 2a and 2b.

Point 3) – not present (jumped to 4)

Point 4) Authors indicate in lines 124-125 about "probably and possibly damaging, were significantly greater in Africans (PolyPhen-2, 14 versus 11 mutations in Europeans, P -value=0.022, two sample t-test)" - is this per patient? please specify.

Response: This is per patient, the word ‘each’ was added for clarity from **Line 126**: “Protein-coding mutations, including probably and possibly damaging, were significantly greater in **each** African (PolyPhen-2, 14 *versus* 11 mutations in a European, P -value=0.022, two-sample t-test; exclusion of hypermutated tumours, P -value = 0.039).”

Point 5) In lines 159-161 it is again noted a difference in prevalence of SPOP coding mutations but the p-value is 0.426.

Response: To minimise the overactions of non-significant findings, we have reduced the text starting at **Main Text Rev1** from **Line 173** to read: “While *TMPRSS2-ERG* fusions (predominantly 3-Mb deletions) were significantly elevated in our European over African tumours (37.7% *versus* 13.3%; OR=3.919, P -value=0.0004), although not significant, African patients were 1.3-fold more likely to present with a *SPOP* coding mutations (MATH and BTB domains).”

Point 6) In Figure 3 there are complete blue bars that extend throughout several gene columns – what does this signify?

Response: These represent genes on Chromosome X considered significant due to homozygous loss. This has been added to the **Figure 3 legend** and reads: “Three genes on chromosome X, *KDM6A*, *ATRX* and *ZMYM3* are considered significant due to the abundance of homozygous loss present in Subtype C.”

Point 7) In Ext Figure 4A, the figure legend doesn’t list the overall cohort numbers or which exact cohort of patients this panel is characterizing. And **Point 8)** In Fig Ext Data 4, it would be helpful to please specify the cohorts being referred to, such as the total number within the group.

Response: Minor addition to **Extended Data Fig 4** included the addition of the ethnicity and number of patients, i.e. Africans (n=113) and Europeans (n=61) labels, as per snapshot below. No changes were made to the figure legend.

Point 9) In Ext Data Figure 5, the authors show a number of affected pathways/genes across the entire group of 183, though it appears some alterations are restricted or enriched in one group; comparisons within the group of significant genes or to other cohorts would be informative.

Response: As addressed under **Referee #1, Major Point 1**, where no other cohorts have ancestry defined data which includes publically available SSM+CN+SV data.

New Extended Data Fig. 5 has been added. The new legend is as follows: “The **search of our 124 significantly mutated genes** is carried out using the TCGA and ICGC cancer databases. The top affected genes for each pathway are present with lollipop plots to show their hotspots of simple coding mutations if they existed. **Mutational frequencies of each altered gene in a pathway are separately measured**

between Africans (n=113) and Europeans (n=61) and shown on the right as a percentage in order (AFR, EUR).”

Point 10) In Fig Ext Data 8 Panel D, the authors state "Copy number and structural variation signatures (CN1-6 and SV1-8, respectively) are the first identified in this study for prostate cancer, and their enrichment in a patient appears to be significantly associated (P-values <0.05) with our GMS." Please clarify how these exact signatures appear and are defined and are associated; how do the authors interpret this?

Response: In the *Figure legend* for *Extended Data Fig. 8*, the following text in red has been added/adapted for clarification. “**d**, Stacked barplots of multiple signature exposures for each mutational type enriched per patient and ranked by ethnic group. **In many cases, certain mutational signatures occur more frequent in a tumour than others. The top enrichment of small- to large-size mutational signatures mentioned is shown for each patient in Supplementary Table 9 (see Enrichment).** Copy number and structural variation signatures (CN1-6 and SV1-8, respectively) are the first identified in this study for prostate cancer, and their **top enrichment of signature mixture/exposure per patient** appears to be significantly associated with our GMS (**one-way ANOVA or Fisher’s exact test, P-values <0.05**), considering either *de novo* or global mutational signatures discovered in the Catalogue of Somatic Mutations in Cancer (COSMIC). **This supports a role of GMS in explaining intrinsic and extrinsic mutational processes in cancer.**”

New Supplementary Table 9 has been provided with Enrichment section.

Point 11) Ext Data Figure 2 which depicts the top 300 genes altered across the cohort was somewhat difficult to interpret given only some gene names listed (TP53 is top but then no label for several rows below it that also show high number of losses); also missing a label of right panel (total counts?). As these could be considered two sub cohorts within the cohort, it could be helpful to also depict comparisons directly between the two cohorts as bar plots.

Response: *Extended Data Fig. 2* was updated for clarification, which included additional text, in red, in the figure legend.

Extended Data Fig. 2 | Somatic driver mutations in 183 prostate cancer patients of different ancestries. The covariates on the right show the total number of altered samples for different mutational types. **a**, Search of the top 300 driver genes altered in primary prostate tumours among 183 specimens. Only driver genes discovered in PCAWG and this study, present in more than six patients or significantly different between Africans and Europeans are chosen for plotting. The top barplot shows the distribution of the number of prostate cancer drivers and/or that of PCAWG. The heatmap shows drivers found in this study (rows) for each patient (columns). Heatmaps are coloured by mutational type. The dual barplot on the left depicts gene-level comparisons of mutational recurrence directly between Africans and Europeans. Bottom covariates show the clinical features of patients. The percentage of transition/transversion mutations across 183 patients shows 1,364,210 small somatic mutations across chromosomes 1-Y. **b**, The bottom heatmap shows the top 22 of previously reported coding driver genes in prostate cancer observed in this study^{7,8,18,19}. The left barplot shows statistical support of recurrence analysis for our study.

Referee #2 (Remarks to the Author):

Comment: This manuscript appears to present findings from the largest and only study of prostate cancer genomes from patients in Sub-Saharan Africa. This is a highly significant and novel study since prostate cancer genomics studies are grossly underrepresented among patients of African ancestry. There are no concerns regarding the approach, quality of data and its presentation. The appropriate statistical tests appear to be used throughout the analysis and conclusions are appropriate.

Response: We thank Referee #2 for recognising not only the novelty and significance of our study, but also for acknowledging the quality of our approach, analysis, and presentation. We will be grateful for the opportunity to present this data to the community and provide prostate cancer genomics a voice within Africa.

Referee #3 (Remarks to the Author):

Comment: In the manuscript by Jaratlerdsiri et al, novel whole-genome sequencing data of prostate tumors is presented that uses somatic variant calling to derive mutational subtypes on a global scale. The major conclusion presented by the evidence in this paper is that the comparison of the subtypes can explain the development of tumors over a timeline comprising the natural life of the patient, due in part to differing mutation rates.

The most novel and compelling component to this paper are the 123 tumor whole-genomes from South African patients. This is itself an amazing resource that is of tremendous value to population scientists and cancer researchers. Strengths of this work include a robust comparison to other datasets (including PCAWG), and the bioinformatic analyses are both well-described and thorough.

Response: Again, we are grateful to Referee #3 for recognising not only the significance of data resource generated and its value beyond only the prostate cancer genomic space, but also for recognising the robust analyses performed, including comparisons with other datasets. Your support is valued by the team.

Comments as Points: Overall, however, I have general and specific concerns that decrease my enthusiasm about the broad conclusions argued.

Response: We would hope that our responses and further clarifications further increase the enthusiasm of this research paper for Referee #3. Addressed point-by-point below.

Point 1) The vast majority of descriptive analyses in the first 2/3 of this work compare the whole-genome analyses of South African patients versus Australian and Brazilian patients. A quick review of Supplementary Table 1 shows significant differences in these groups, including the use of a biopsy sample (SA) vs prostatectomy (BR and AU) and their baseline PSA levels (239 vs 15). This alone argues that the SA cohort is potentially higher risk, and if any subset of these patients had metastatic disease (which is highly likely) the age/timing of the disease is different as is the possibility of metastatic re-seeding that could influence downstream analyses. If any of these patients had metastatic disease the conclusions of this work would be different.

Response: As discussed extensively under *Referee #1, Point 6*, while one may assume the South African cohort represents patients at higher prostate cancer risk, based on Gleason scores, the South African cohort presented with a lower percentage of high-grade (ISUP GG 4/5) tumours (72.2%), than the Australian (86.6%) and the smaller Brazilian (85.7%) cohorts. The risk of metastatic re-seeding would therefore be arguably slightly higher in the Australian than the South African cohort. As follow-up data was available for the Australian and the Brazilian cohorts, we are aware that 21/53 Australians would eventually present with bone (majority), distant visceral or nodal metastasises, while 3/53 would

eventually present with local pelvic metastasis. Four of the Brazilians reported bone or lymph node metastasis, one died and two had unknown follow-up data.

We have previously reported [1] high baseline PSA levels in Black South African (via the SAPCS) men through analysis of 917 participants, including 576 clinicopathological confirmed prostate cancer cases and 341 controls (no detectable prostate cancer presenting at a participating urology clinic). Of the cases 83.2% presented with a PSA ≥ 20 ng/mL (median 98.8 ng/mL), observing a significant prostate cancer risk with a PSA ≥ 10 ng/mL vs. < 4 ng/mL ($p < 0.0001$). Cases with a PSA level ≥ 100 ng/mL were at an increased risk of presenting with a poorly differentiated tumor (RR=6.84 95%, CI 1.28-36.56, $p=0.0246$), while 36% of cases presented with high-risk prostate cancer defined as per this study as Gleason score > 7 . Of note, less than a quarter of the controls presented with a PSA below the global cut-off of 4 ng/mL, 28.8% with a PSA ≥ 20 ng/mL (median 9.1 ng/mL). Compared with the US-based SEER (surveillance Epidemiology & End Results) registry data for African Americans, SAPCS men showed elevated PSA levels (17.2% vs. 83.2% PSA ≥ 20 $\mu\text{g/L}$) at presentation. Critically, 81.5% of the SAPCS pathologically confirmed controls presented with a PSA < 20 $\mu\text{g/L}$. While SAPCS men were more likely to present 2-5 years later than global studies, the proportion of SAPCS cases < 70 years presenting with high-risk prostate cancer was significantly greater than African Americans matched for age and tumor pathology. It is therefore our argument that men from Africa present with higher PSA levels which should not necessarily be equated to global norms. Scanning the literature for additional evidence we note elevated PSA reported for 543 cases diagnosed between 2005 and 2016 in Central Sudan [2], with PSA levels > 4 , > 20 and > 100 ng/mL reported for 91%, 77% and 53% of the cases, respectively, while our colleagues in Kenya (unpublished data) report for 975 cases between 2009 and 2019, PSA levels > 100 ng/mL for 57.4% of cases. Studies are underway to generate a new classification of PSA levels applicable for African men. **References:** (1) Tindall EA, *et al*, Clinical presentation of prostate cancer in black South Africans. *Prostate* 2014; 74:880-891. (2) Taha et al. Prostate cancer clinical characteristics and outcomes in Central Sudan. *edicalscience*. 2020; 14:1116.

We therefore do not believe that the South African cohort is unmatched to the Australian (and Brazilian) cohort with regards to their potential to have already undergone metastatic seeding or re-seeding at time of diagnosis/surgery, while we acknowledge the unavoidable (due to lack of routine PSA screening) overall later age at presentation of the South African cohort.

Point 2) Despite these clinical differences, the objective mutational profiles between their cohorts (as presented in Figures 1 and 2) do not show much distinction between geographic groups. Even when limiting to drivers from unbiased analyses, integrative clustering does not demonstrate compelling differences. While this still is in alignment with the manuscript's goal of presenting a molecular taxonomy, it does not change the current views of prostate cancer from previous datasets.

Response: We would like to re-emphasise the elevated Tumour Mutational Burden (TMB) and Percentage genome Alteration (PGA) observed for the African *versus* European derived tumours, both of which were significant, see from **Line 109**, where TMB remained significantly elevated in Africans even when the hyper-mutated tumours were excluded. We further refer to **Referee #1 Major Point 1** and **Point 2 responses** where we highlight ethnic differences in both oncogenic drivers (new **Figure 2a and 2b**), while in new **Figure 3a**, we would argue that the African-specific GMS-B shows compelling profile differences defined by Copy Number (CN) gains. Furthermore, as highlighted in new **Figure 4b**, we show duplication Signatures CN4 (whole genome duplication) and SV5 (large duplication) to be significantly over-represented in African-specific GMS-B.

Point 3) Some discussion of alterations and driver genes are confusing. Data presented in the text and in Figure 3 suggest that some alterations occur in 100% of samples (line 173). Please consider editing the methods to validate or curate these striking findings.

Response: *Figure 3* has been updated to avoid confusion, see **Referee #1, Point 2 Response**. The confusion with the alterations in 100% of samples is a direct result of the single X chromosome in men. In *Figure 3*, top panel, the 100% representation for the genes on X chromosome have been removed, while these genes have been further highlighted (boxed) in the heatmap for further clarity.

Point 4) The initial presentation of the GMS classifier is itself confusing. Is it supposed to be presented in *Figure 3b*? (The manuscript says 2b.) Are each group of rows/boxes the GMS A-D? Consequently, interpretation of data based on the GMS scores (including the SBS signatures) do not reference individual GMS groups, so I struggled to appreciate the significance of the data presented in *Figure 4* and how it related to these GMS groups.

Response: *Figure 3* and *Figure 4* have been updated for clarity, see **new Figure 3** and **new Figure 4**, the latter further highlighting the significance of the GMS groups.

Point 5) The molecular evolution and timing data presented in *Figure 5* is quite strong and strengthened by the Extended data. However, the confidence intervals around mutation rates (5a and 5b) are extremely large with respect to patient age. To what extent could these differences be driven by the higher risk clinical features of the South African group?

Response: Addressed extensively under **Point 1 Response**, as well as **Referee #1 Point 6 Response**.

Minor concerns:

Point 1) The paper refers to their cohort as Sub-Saharan, African and South African. Supp Table 1 indicates all patients were from South Africa, the country. As Sub-Saharan Africa comprises over 40 countries, this language should be clarified.

Response: Within the context of the abstract and introductory paragraph we highlight the lack of data for **Sub-Saharan Africa**, with our study being the largest of its kind for the region. We believe that, over 20 years since the completion of the Human Genome Reference, it is important to highlight the lack of genomic data and in particular cancer genomic data for this large region, which is further significant due to the exasperated prostate cancer mortality rates. Throughout the paper we refer to the geographic identifier as **South Africa** (this also includes 5 European South Africans and 5 genetically Admixed South Africans), while we define, through genetic ancestry analysis the ethnicity of the cohort as **African** (including 113/123 South Africans). We have gone through the text to ensure that there is no confusion related to this terminology, making changes (red) to enhance clarification. We explain further under **Genetic ancestries** how African (>98% African ancestry) and European (allowing for minimal Asian ancestry yet excluding for African ancestry) ancestry has been defined in the context of our study. From this point onwards patients are largely referred by their genetic ancestry as African or European, while using Europeans are further classified by geography as Australian, South African or Brazilian.

Point 2) Some of the intervals presented are non-sensical. For example, Line 127 says a median of 2+/- 22.5 coding drivers was observed. Since -20.5 coding drivers cannot occur, perhaps a different two-sided analysis should be used accepting a distribution that is left-censored at zero?

Response: For clarity the sentence has been adapted, text in red, to read in the **Main Text Rev1** from **Line 131** as “A median of **two (Q1-Q3, 2-4)** coding drivers was observed in this study (Supplementary Table 2), with **one (0-2)** appearing to be prostate cancer-specific.

Referee #4 (Remarks to the Author):

Comment: Hayes and colleagues conduct an important study in the WGS of sub-Saharan African genomes of men with prostate cancer and compare these to non-African prostate cancers. This is the first study which addresses African continent prostate cancers which may of course be different from African American or Afro-Caribbean prostate cancers-studies the authors argue for in later studies. They point out important differences in DNA mutations, cancer drivers and come up with a novel taxonomy that groups their cohort into 4 groups-one that is highly ethnically mixed and 2 that appear to be African specific. They further use clonal evolution analyses to associate clock-like processes to mutation acquisition and infer differential causality based on signature analyses in the African genomes.

The work is novel. The implications for prostate cancer etiology and personalised medicine approaches could be very important.

Response: We thank Referee #4 for acknowledging the significance of this work, for highlighting the novelty and importance of African inclusion, while raising the relevance for clinical impact. We appreciate your support.

Comment: My major initial concern is the initial cohort itself in the comparison between the caucasian and African clinic-pathologic characteristics. Extended Figure 1a should be modified and statistically compared between African and non-African patients such that there is not bias in the characteristics between the two groups. If the clinical factors are more aggressive in the African group, then relative already- known entities such as PGA, mutations, drivers etc can be explained by higher GG or PSA or T-category or intraductal carcinoma. This is less of an issue when there are novel findings for African genomes, but very important for the relative calling of mutations of African vs non-African.

Response: As presented in response to ***Referee #1 (Point 6), Referee #3 (Point 1)***, we show that the South African cohort does not present with a higher overall ISUP Grade Grouping than the Australian or small Brazilian cohort. As highlighted in ***new Extended Data Fig.1a***, we show, Europeans to have significantly higher ISUP (IQR, GG4-5) than Africans (IQR, GG3-5). As with this study, we and others (although limited data for Africa exists) have reported exceptionally higher PSA levels among Africans, irrespective of their cancer status, see comments under Referee #3 Point 6 where we highlight these studies. We are therefore of the strong opinion that western criteria for associating PSA levels with prostate cancer status is not applicable within greater Sub-Saharan Africa and much work needs to be done to establish a new set of criteria.

New Extended Data Figure 1 adapted to include significant differences between the cohort as described in red in the figure legend.

a

	African ancestry	European ancestry	Admixed ancestry
Country of origin	113 patients**	61 patients**	9 patients**
South Africa	113	5	5
Australia	0	53	0
Brazil	0	3	4
Clinical age (median, Q1-Q3)	68 years (63-71)**	63 years (58-67)**	64 years (59-71)
45 - 60 years	21	22	3
61 - 75 years	77	38	5
≥ 76 years	14	1	1
Pre-operative PSA (median, Q1-Q3)	52.2 ng/ml (24-133)**,*	8.2 ng/ml (6-12)**	17.4 ng/ml (10-79)*
< 10 ng/ml	9	40	2
10 - 20 ng/ml	12	17	3
≥ 20 ng/ml	86	4	4
ISUP grade group (median, Q1-Q3)	GG 4 (GG 3-5)**	GG 5 (GG 4-5)**	GG 4 (GG 4-5)
GG 0 - 2	19	8	2
GG 3	10	1	0
GG 4	38	11	3
GG 5	38	41	4
WGS tumour purity (median, Q1-Q3)	48 % (42-52 %)**	45 % (39-56 %)**	44 % (42-45 %)
10 - 40 %	24	23	1
41 - 60 %	80	26	8
61 - 90 %	9	12	0

Extended Data Fig. 1 | Clinical cohorts and statistical metrics. **a**, Clinical and pathological patient characterisation. Pairwise comparisons using contingency tables and Fisher's Exact test between African ancestry and Admixed/European ancestry are highlighted in bold with P -value <0.05 (*) or <0.01 (**). Summary statistics, including the median, first and third quartiles (Q1-Q3), are also present. **b**, STRUCTURE analysis of bi-allelic germline variants with the logistic prior model. Model components used to explain structure in the plot are $K=5$. All spectrum of African contributions are summed and assigned as African ancestry. **c**, Saturation curve for all driver types across 183 patients. Recurrent copy number gains and losses were measured using GISTIC v2 (Supplementary Methods). CDS, coding sequence; SV, structural variation. **d**, Spearman's correlation between different variables measured in this cohort. Dot sizes represent the magnitude of correlation, with significant P -values <0.01 .

Comment: In some instances, the authors describe "significant" increases in mutations in African genomes with supporting p-values, but it can be perceived as slightly disingenuous to say "increased" with a non-significant p-value. Even if an absolute value is larger, the authors should rather state "no difference" if the statistics do not support a difference. Otherwise, there is a subconscious bias in the reader that more differences exist than actual testing supports.

Response: Areas where non-significant findings were reported have either been removed or sentences altered to reflect the lack of significance more accurately. Please refer to *Referee #1, Minor Point 1* and *Minor Point 5 Responses* for details.

Comment: A small point is the use of the PGA values - this is somewhat confusing - can the authors use percentages for PGA (e.g. 0.078; is this not 7.8% rather than 0.078%? The reader could be confused relative to PGAs from 1-30% in localised disease reported in other papers.

Response: This has been updated in *Supplementary Table 1*. Additionally, from *Line 179* percentages were added and reads: We found Subtype A to be mutationally quiet (1.01 mutations/Mb, 0.50 breakpoints/10Mb, 2% PGA); conversely Subtype D showed the greatest mutational density (1.91 mutations/Mb, 1.08 breakpoints/10Mb, 31% PGA).

Comment: Can the authors be more explicit in terms of differences between African versus African-American genomes? There must be some WGS carried out-certainly there are mutation spectra in AACR GENIE that have been published in NEJM between African-American, Caucasian and Asian populations-this might strengthen their case.

Response: A comparative analysis was performed between AACR GENIE and our study, which has been included in the *Supplementary Methods, Section 8.4.3*. In summary, of the 865 primary prostate cancer genomes, 56 were derived from African Americans. The most significant limitation in comparing the AACR GENIE and OUR STUDY include: (i) FFPE versus fresh prostate tissue, and (ii) gene targets versus whole genome. See added text below and added Supplementary Figure providing the analysis.

8.4.3 AACR GENIE

The AACR Project Genomics, Evidence, Neoplasia, Information, Exchange (GENIE) consists of 865 patients with primary prostate cancers (FFPE) and accompanying clinical data including their primary race of Asian, Black or White (n=20, 56 and 789, respectively). All the patients have the following three types of cancer genomic data available for exonic regions: *i*) CDS mutations (183 genes); *ii*) gene fusions (183 genes); and *iii*) gene-level copy number alterations by GISTIC (176 genes). The patients are from two American cohorts, DFCI and MSK. The AACR GENIE data (version 11.0-public) were retrieved via Synapse Platform (<https://synapse.org/genie>) [1, 2]. For comparisons, the same gene panels were extracted from CPGA and our cohort and all the three cohorts proceeded with integrative clustering analysis (Section 8.1). Visualisation of prostate cancer subtypes is illustrated in Supplementary Figure X, with GMS-A, -B, -C and -E present simultaneously with the proportion of altered genes across the panel examined per patient for the three mutational types. GMS-D originally identified in our study using whole genome data appears to be combined to GMS-E.

Supplementary Figure 5. Integrative clustering analysis of AACR GENIE patients with primary prostate tumours reveals two distinct molecular subtypes. The concurrent analysis with CPGEA and our cohort shows four subtypes, with GMS-D missing. The AACR GENIE includes African Americans identified as Black. Note that the level of genetic admixture among AACR GENIE samples, especially African Americans are unknown.

Comment: The argument around SBS, ID and other signatures in African versus non-African and across GMS groups A-D is unclear in the manuscript-what is the interpretation regarding etiology of the subgroups when this is applied.

Response: See *Referee #1, Point 3 Response*, which includes the addition of *Fig. 4b* and the added text from *Line 276* which reads: “*Fig. 4b* shows the duplication signatures to have at least 1.5 times larger proportion of genomic aberrations in GMS-B, -C and -D than GMS-A. GMS-B appeared to be the African ancestry-specific subtype in this study also consisted of several CN4 and SV5 genomic aberrations composed predominantly of CN amplification (>5 copies and mainly >100 kb in length) and tandem duplication (<5 Mb in size occurred during early to late replication timing), respectively.” Additionally, from *Line 285*, we removed the following sentence to avoid ambiguity, which read: “Small-size signatures were inversely significant among 20 mutated genes, indicating a higher number of mutations towards lesser mutated tumours”.

Comment: Can the authors be explicit surrounding the timing of the chromosomal gains and losses within African versus non-African genomes?

Response: An additional panel (e) was added to *Extended Data Fig.10*, summarising the timing of chromosomal gains and losses between the ethnicities, with text added to the figure legend as indicated in red.

New Extended Data Fig.10e

Text added to figure legend: **e**, Molecular timing distribution of copy number gains and loss of heterozygosity (LOH) between Africans and Europeans. Pie charts depict the distribution of the inferred mutation time for a given copy number alteration. Orange denotes early clonal gains/LOH, with a gradient to green for late gains/LOH. The size of each chart is proportional to the recurrence of this event across different patients. Most of the gains and LOH are considered early clonal based on MutationTimeR results. Whole genome duplication is more frequent in Africans (63%) than in Europeans (57%).

Comment: The pathway analyses are important and do relate to the TCGA pathways published, but the gene-specific reasons for the pathways are non clear. For example, when showing data in which DDR -genetic instability is increased in African versus non-African, what genes or processes are driving this ? Is it mismatch repair defects as evidenced from the signature analyses (as the HR signatures do not seem to be as important) or specific genes - I note that this study would suggest that BRCA2 is not a driver in Africa? Is that the case?

Response: So far, we can only identify “DNA repair/Genome integrity” from our 124 preferentially mutated genes among subtypes using maftools and its collection of pathway genes from ICGC/TCGA, but can’t provide detailed analysis of DDR classes, as only 2/124 are DDR genes for pathway analysis. See **Extended Data Fig. 5** for *TP53* (considered DDR by ICGC/TCGA) and *ATM*, with *TP53* mutated in 52% of African and 40% of European derived tumours. While driver BRCA1/2 variants were not identified in our study, one cannot exclude the potential significance of BRCA2 in Africa, we would argue that larger studies, capturing further African geo-ethnic diversity would be required to make any conclusions regarding which genes are not drivers in Africa. While not the focus of this study, both germline BRCA1/2 predicted deleterious germline mutations were identified in our African cohort and are part of a manuscript under review. From the signature analysis SBS3 was rare across the tumours, while Signatures such as SV4 (small deletion), SV7 (small tandem duplication & translocation) and ID83F (deleted microhomology) might explain HR.

Comment: For clarity and context to a generalist audience for Nature or Nature journals, the abstract should be placed into more of a lay context.

Response: Redrafted ABSTRACT: Prostate cancer is characterised by significant **geo-ethnic** disparity. **With African ancestry a well-established risk factor, mortality rates across Sub-Saharan Africa are double European and quadruple Asian countries¹. The contributing genetic and non-genetic factors leaving a characteristic somatic mutational pattern are unclear within this cancer^{2,3}. Here, through whole-genome sequencing of treatment-naïve prostate cancer from 183 ethnically/globally distinct patients (African versus European), we generate the largest cancer genomics resource for Sub-Saharan Africa, identifying ~2 million somatic variants. Interrogating all somatic mutational types merged with geo-ethnic identifiers, we describe a new prostate cancer molecular taxonomy. Defined as Global Mutational Subtypes (GMS), while Africans present within all subtypes, we found GMS-B characterised by copy number gains and mutationally-noisy GMS-D to be African/Africa-specific. Including Chinese Asian data, we show mutationally-quiet GMS-A to be geo-ethnically universal, while African-European restricted GMS-C characterised by copy number losses predicts poor clinical outcomes. Significant African-specific findings include an elevated tumour mutational burden, percentage genome alteration, number of predicted damaging mutations and total mutational signatures, and driver genes *NCOA2*, *STK19*, *DDX11L1*, *PCAT1* and *SETBP1*. Besides the clinical benefit of African inclusion, our GMS reveal different evolutionary trajectories and mutational processes, suggesting both common genetic and environmental factors are contributing to the disparity. [201 words][Perspective] Analogous to gene-environment interaction defined here as a different effect of an environmental surrounding in persons with different ancestries or vice versa, we anticipate GMS acting as a proxy to intrinsic and extrinsic mutational processes in cancers, promoting global inclusion in landmark studies. [201+43 words = 244 words]**

Comment: A summary table at the end of the manuscript pointing out the major differences between African and non-African prostate genomes would be very helpful and I suspect off quoted and shown in many slide sets. I

Response: *Supplementary Table S12* has been added and references in the *Main Text Rev1*, from *Line 315*, “To our knowledge, our study represents the first, if not, the largest whole-genome prostate cancer, and likely any cancer, genome resource for Sub-Saharan Africa (see summary in *Supplementary Table 12*).”

Added references

1. Mahal BA, Alshalalfa M, Kensler KH, Chowdhury-Paulino I, Kantoff P, Mucci LA, Schaeffer EM, Spratt D, Yamoah K, Nguyen PL *et al*: **Racial Differences in Genomic Profiling of Prostate Cancer**. *N Engl J Med* 2020, **383**:1083-1085.
2. AACR-Project-GENIE-Consortium: **AACR Project GENIE: powering precision medicine through an international consortium**. *Cancer Discov* 2017, **7**:818-831.

Reviewer Reports on the First Revision:

Referees' comments:

Referee #3 (Remarks to the Author):

I have read the authors' response to my and the other reviewers' concerns and I commend the authors on their efforts to improve the clarity and presentation of the data.

With respect to my first point, which concerned the baseline higher PSA levels for SA patients vs their European counterparts, I have read the paper the authors indicated (which I believe is Tindall et al Prostate 2014, not Mahal et al NEJM 2020). The shift towards higher PSA reported in the "Prostate" manuscript is striking. In that manuscript, the authors discuss at length stage migration associated with higher PSA and age of diagnosis, even when adjusting for age. I do not think I will be the only prostate cancer researcher who has a similar question when reviewing the data in this current manuscript, and as such, I feel that the authors should address this head-on, at least briefly. For example, I scrutinized both the original and revised manuscript (and supplement) to identify some mention of the differing median PSA's between cohorts, and found none. I similarly did not find any discussion of patient clinical stage ("TNM" or equivalent) or the presence/absence of metastasis at the time of diagnosis -- which would indicate whether PET/CT and/or bone scans were performed to rule out the presence of metastasis in either cohort prior to biopsy (SA) or prostatectomy (controls). If both cohorts are comprised of patients with truly locally advanced disease (that is, no evidence of metastasis after appropriate staging), please indicate this. If staging was not performed, please indicate this as well and specify it as a limitation in interpreting the results.

Referee #4 (Remarks to the Author):

I thank the authors for addressing all of my concerns.

The new data and new representations of the data are compelling and the manuscript reads much better overall and now conveys the importance and novelty of their study.

Author Rebuttals to First Revision:

Referees' comments:

Referee #3 (Remarks to the Author):

I have read the authors' response to my and the other reviewers' concerns and I commend the authors on their efforts to improve the clarity and presentation of the data.

Response: We appreciate the feedback and contribution made by the reviewers.

With respect to my first point, which concerned the baseline higher PSA levels for SA patients vs their European counterparts, I have read the paper the authors indicated (which I believe is Tindall et al Prostate 2014, not Mahal et al NEJM 2020). The shift towards higher PSA reported in the "Prostate" manuscript is striking. In that manuscript, the authors discuss at length stage migration associated with higher PSA and age of diagnosis, even when adjusting for age. I do not think I will be the only prostate cancer researcher who has a similar question when reviewing the data in this current manuscript, and as such, I feel that the authors should address this head-on, at least briefly. For example, I scrutinized both the original and revised manuscript (and supplement) to identify some mention of the differing median PSA's between cohorts, and found none. I similarly did not find any discussion of patient clinical stage ("TNM" or equivalent) or the presence/absence of metastasis at the time of diagnosis -- which would indicate whether PET/CT and/or bone scans were performed to rule

out the presence of metastasis in either cohort prior to biopsy (SA) or prostatectomy (controls). If both cohorts are comprised of patients with truly locally advanced disease (that is, no evidence of metastasis after appropriate staging), please indicate this. If staging was not performed, please indicate this as well and specify it as a limitation in interpreting the results.

Response: The reviewer is correct, the reference is Tindall et al., 2014. There is also no clinical data available for the South African cohort (sampled at diagnosis), while there is data for the Australian cohort (see below), all metastatic free at time of surgery. For further clarification related to the differences in clinical presentation between the cohorts, we add the following text in red.

Main text; from Line 68: Through the Southern African Prostate Cancer Study (SAPCS), we report a 2.1-fold increase in aggressive disease (Grades 4-5) and 4.8-fold increase in prostate specific antigen (PSA) levels at diagnosis compared to African Americans¹⁷.

Main text, from line 75: Focusing on treatment-naïve cases (100% South Africans, 98% Australians, and two confirmed Brazilians) and aggressive tumours (Grades 4-5 for 72.2% South Africans, 86.8% Australians, and 85.7% Brazilians; Extended Data Fig. 1a) at biopsy (100% South Africans) or surgery (100% Australians, 62.5% Brazilians) and patient-matched blood achieving coverages of 88.69±14.78 and 44.34±8.11, respectively (median±s.d., Supplementary Table 1), we uniformly generated, called and assessed about 2 million somatic variants.

Main text, Discussion, from Line 319: Appreciating the lack of clinical staging for the South African patients (recruited at diagnosis), we describe a novel prostate cancer molecular taxonomy, identifying ancestrally distinctive Global Mutational Subtypes (GMS).

Methods, Patient cohorts and whole-genome sequencing: Our study included 183 treatment naïve prostate cancer patients recruited under informed consent and appropriate ethics approval (Supplementary Methods, Section 2) from Australia (n=53), Brazil (n=7) and South Africa (n=123). While matched for pathological grading, as previously reported¹⁷, PSA levels are notably elevated within our African patients, while we cannot exclude for potential metastasis. DNA extracted from fresh tissue and matched blood underwent 2x150 bp sequencing on the Illumina NovaSeq instrument (Kinghorn Centre for Clinical Genomics, Garvan Institute of Medical Research).

Supplementary Table S1: Clinical Stage has been provided in Table S1 for the Australians.

Referee #4 (Remarks to the Author):

I thank the authors for addressing all of my concerns.

The new data and new representations of the data are compelling, and the manuscript reads much better overall and now conveys the importance and novelty of their study.

Response: The reviewers' comments and appreciation for our revision is warmly received.